# Mixed Sparsity Training:
# Achieving 4× FLOP Reduction for Transformer Pretraining

**Pihe Hu**[*]                                                                *hupihe@gmail.com*
*Institute for Interdisciplinary Information Sciences*
*Tsinghua University*

**Shaolong Li**[*]                                                          *shaolongli16@gmail.com*
*Central South University*

**Xun Wang**                                                    *wang-x24@mails.tsinghua.edu.cn*
*Institute for Interdisciplinary Information Sciences*
*Tsinghua University*

**Longbo Huang**[†]                                                *longbohuang@tsinghua.edu.cn*
*Institute for Interdisciplinary Information Sciences*
*Tsinghua University*

**Reviewed on OpenReview:** *https://openreview.net/forum?id=XosdLS7KVE*

## Abstract

Large language models (LLMs) have made significant strides in complex tasks, yet their widespread adoption is impeded by substantial computational demands. With hundreds of billion parameters, transformer-based LLMs necessitate months of pretraining across a high-end GPU cluster. However, this paper reveals a compelling finding: *transformers exhibit considerable redundancy in pretraining computations*, which motivates our proposed solution, Mixed Sparsity Training (MST), an efficient pretraining method that can reduce about 75% of Floating Point Operations (FLOPs) while maintaining performance. MST integrates dynamic sparse training (DST) with Sparsity Variation (SV) and Hybrid Sparse Attention (HSA) during pretraining, involving three distinct phases: warm-up, ultra-sparsification, and restoration. The warm-up phase transforms the dense model into a sparse one, and the restoration phase reinstates connections. Throughout these phases, the model is trained with a dynamically evolving sparse topology and an HSA mechanism to maintain performance and minimize training FLOPs concurrently. Our experiment on GPT-2 showcases a FLOP reduction of 4× without compromising performance.

## 1 Introduction

Over the past years, the field of Large Language Models (LLMs) has witnessed remarkable advancements, e.g., T5 (Raffel et al., 2020), GPT-3 (Brown et al., 2020), GLM (Du et al., 2021). These sophisticated models, characterized by their vast scale, typically with $1 \sim 200$ billion parameters, have redefined the frontiers of language understanding, generation, and comprehension. Their prowess in tasks spanning from question-answering systems (Liu et al., 2023), programming tools (Roziere et al., 2023) to video generation (Bai et al., 2023), has propelled them to the forefront of research and applications in many fields.

However, the realization of their potential comes entwined with an imposing bottleneck: the substantial computational cost required for their pretraining. The process of priming these colossal models necessitates

---

[*]Equal contribution.
[†]Corresponding author.

extensive computational resources (Chen et al., 2023a), involving protracted periods spanning several months running on clusters comprising thousands of high-performance Graphics Processing Units (GPUs). For example, Narayanan et al. (2021) point out that GPT-3 175B (Brown et al., 2020) requires about 34 training days on a cluster of 1024 NVIDIA A100 GPUs, and Llama-2 70B (Touvron et al., 2023) even requires double training time on the same cluster.

The substantial computational overhead associated with pretraining transformer-based LLMs stands as a formidable barrier, restricting their widespread adoption and accessibility. While existing efforts have sought to enhance pretraining efficiency through strategies such as training parallelism (Li et al., 2020; Bian et al., 2021; Zhao et al., 2023), hardware-assisted attention operators (Dao et al., 2022c; Dao, 2023), and mixed precision training (Micikevicius et al., 2017; Burgess et al., 2019; Liu et al., 2022), these methods primarily address system-execution-level bottlenecks without tackling the intensive algorithm-level FLOPs associated with training. In essence, the algorithmic computation redundancy of transformer pretraining remains an understudied domain. Furthermore, traditional FLOP reduction techniques, including pruning (Ma et al., 2023; Frantar & Alistarh, 2023; Syed et al., 2023) and quantization (Dettmers et al., 2022; Frantar et al., 2022; Frantar & Alistarh, 2022), have demonstrated the presence of significant FLOP and parameter redundancy within transformers. However, these approaches are primarily applicable in post-training stages, rendering them impractical for addressing computational costs during the pretraining of transformers. This emphasizes the critical need for mitigating algorithmic inefficiencies inherent in the pretraining phase of transformers.

Take GPT-2-small (Brown et al., 2020) as an example. Figure 1 illustrates that fully-connected layers (Basha et al., 2020) and self-attention layers (Vaswani et al., 2017) contribute significantly to the FLOPs, accounting for up to 83% and 15%, respectively. The FLOPs of fully connected layers stem from weight updates, while the computation-intensive self-attention operations drive the FLOPs of self-attention layers. Given the distinct characteristics of these layers, different methods are required to reduce training FLOPs. Dynamic Sparse Training (DST) (Mocanu et al., 2018; Evci et al., 2020) has emerged as an effective solution to reduce the FLOPs of fully connected layers, demonstrating the capability to train 80%-sparse networks such as ResNet-50 (He et al., 2016) and MobileNet (Howard et al., 2017) from scratch without performance degradation. However, direct application of DST methods in pretraining transformers, as seen in SET (Mocanu et al., 2018) for BERT (Dietrich et al., 2021), only yields modest FLOP savings, typically less than 20%, to maintain performance, owing to the extensive parameters required by transformers for complex tasks.

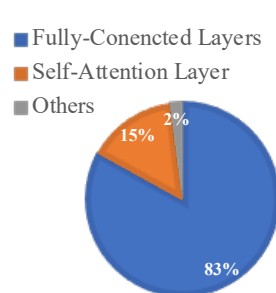

Figure 1: Pretraining FLOPs of GPT-2, detailed in Appendix A.5.

Another methodology to reduce training FLOPs of fully-connected layers is to utilize block sparse matrix, such as Dao et al. (2022a;b), yet the maximum FLOP reduction is incomparable to our MST methods as shown in Table 1, since they adopt a fixed sparse pattern whose sparsity level is limited to maintain the performance. Meanwhile, sparse attention mechanisms (Child et al., 2019; Dai et al., 2019) have been proposed to reduce the FLOPs associated with attention operations. However, the application of sparse attention alone typically results in savings of non-dominant FLOPs, as depicted in Figure 1. These observations prompt the following open question: ***Can we remove the computational redundancy in transformer pretraining without losing performance?***

In response to the formidable challenge of computational overhead in transformer pretraining, this paper introduces a novel method called Mixed Sparsity Training (MST). Unlike existing approaches, MST seamlessly integrates dynamic sparse training with Sparsity Variation (SV) and Hybrid Sparse Attention (HSA) throughout the pretraining process, unfolding in three pivotal phases: warm-up, ultra-sparsification, and restoration. As shown in Figure 2, the warm-up phase transforms the dense model into an initial sparse topology, addressing uncertainties in link importance at the initial training stage. Subsequently, the deployment of a novel sparse training method, Mixed-Growing (MG), facilitates the simultaneous training of a sparse model while actively exploring other sparse topologies. The restoration phase then intelligently reinstates connections, effectively recovering performance loss encountered during the ultra-sparsification phase. Additionally, MST introduces HSA to mitigate FLOPs generated by self-attention layers, by incorporating

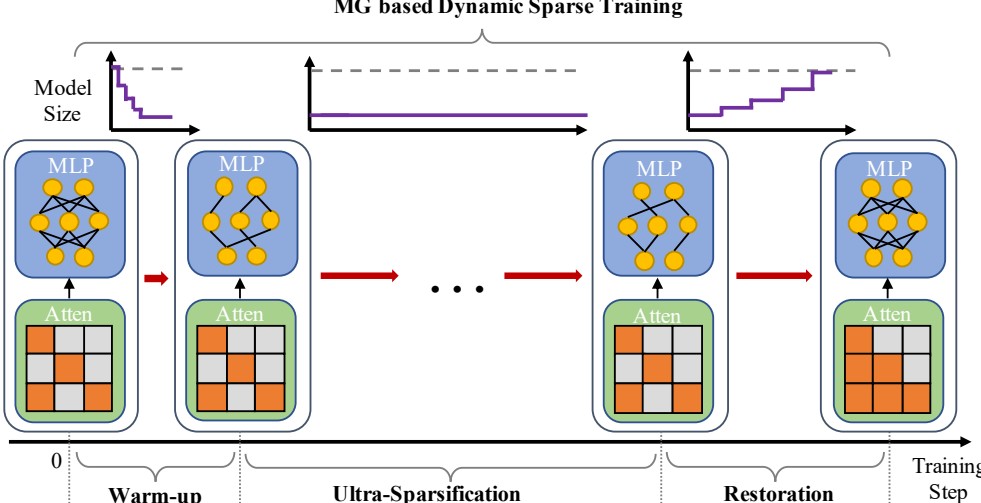

Figure 2: The sparsity variation of MST includes three phases: warm-up, ultra-sparsification and restoration. SV is combined with MG-based dynamic sparse training and HSA during the training.

sparse attention mask in the first two phases, and then transitioning to a densified attention mask when the network weights are dense.

In essence, MST stands poised to revolutionize transformer pretraining, offering significant FLOP savings while providing users with a transparent and efficient method for training dense transformers. Our experiments with MST on GPT-2 demonstrate an exceptional FLOP reduction of 4× without compromise in performance across multiple zero-shot and few-shot downstream tasks. Moreover, MST is entirely orthogonal and can seamlessly integrate with existing system-level acceleration methods, such as training parallelism (Zhao et al., 2023), hardware-assisted attention operators (Dao, 2023), and mixed precision training (Liu et al., 2022), thus facilitating efficient transformer pretraining and achieving higher acceleration ratios. Table 1 provides a comprehensive summary of related works aiming to reduce pretraining FLOPs, highlighting MST's state-of-the-art performance in this regard. A detailed literature review is provided in Appendix 2.

Table 1: Comparison of different training techniques for saving transformer pretraining FLOPS.

| Name | Methods | Models | Reduction |
|---|---|---|---|
| DynSparse (Dietrich et al., 2021) | Dynamic Sparsification | BERT | 2× |
| LiGO (Wang et al., 2023) | Layer Growth | BERT | 2.2× |
| Monarch (Dao et al., 2022b) | Block Sparsification | GPT-2, BERT | 2.2× |
| SPDF (Thangarasa et al., 2023) | Static Sparsification | GPT-2, GPT-3 | 2.5× |
| Pixelated Butterfly (Dao et al., 2022a) | Butterfly Sparsification | GPT-2 | 2.6× |
| **MST** (Ours) | Dynamic Sparsification | GPT-2 | 4× |

## 2 Related Work

We include most related works about dynamic sparse training and transformer pruning in this section.

### 2.1 Dynamic Sparse Training

Dynamic sparse training (DST) is developed to reduce both computational workload and memory usage throughout the entire training process. It involves training a sparse neural network from scratch while dynamically adjusting and updating the sparse mask during training. Sparse Evolutionary Training (SET) (Mocanu et al., 2018) removes least magnitude weights and reintroduces random weights for better exploration at the end of each training epoch. SNFS (Dettmers & Zettlemoyer, 2019) utilizes exponentially smoothed momentum as the criterion for weight growth. RigL (Evci et al., 2020) uses the same magnitude-based pruning method while activating new weights based on their gradient magnitudes through infrequent full gradient calculation, leading to improved accuracy under the same sparsity.

However, the greedy-based growth policy is highly likely to result in limited weight coverage, consequently yielding a sub-optimal sparse structure network. ITOP (Liu et al., 2021b) delves into the fundamental mechanism of DST, revealing that the advantages of DST arise from exploring all potential parameters over time. Top-KAST (Jayakumar et al., 2020) prunes least magnitude weights, but it updates a superset of active weights determined by backward sparsity to enable the revival of pruned weights. To achieve high-quality results, more weight gradients need to be involved in the computation throughout the training, which means there is less backward sparsity and a substantial increase in computational workload. Schwarz et al. (2021) proposed a weight-parameterization that keeps the low-magnitude parameters unaffected by learning largely, and its application contributes to enhanced accuracy within the Top-KAST framework. AC/DC (Peste et al., 2021) employs co-training to simultaneously train both sparse and dense models, allowing for a flexible transition between sparse and dense training and yielding accurate results for both sparse and dense models. MEST (Yuan et al., 2021) adopts a gradually decreasing drop and grow rate during training and introduces a memory-efficient training framework designed for fast execution on edge devices. Spartan (Tai et al., 2022) integrates soft masking with dual averaging-based updates through a computationally expensive process of calculating a soft top-k mask.

### 2.2 Transformer Pruning

Parameter pruning can certainly be applied to the compression of transformers as well by eliminating redundant model weights. The methods for parameter pruning in transformers fall into three main categories: structured, semi-structured and unstructured pruning.

**Structured pruning.** Structured pruning concentrates on removing larger structured patterns of the network such as groups of consecutive parameters or hierarchical structures like rows, columns, or sub-blocks of the transformer weight matrices, resulting in a model that does not require specific hardware or software for acceleration. Globally Unique Movement (GUM) (Santacroce et al., 2023) prunes network components based on their global movement and local uniqueness scores, aiming to maximize both sensitivity and uniqueness. LLM-Pruner (Ma et al., 2023) employs gradient information to selectively remove non-essential interconnected structures. It computes the structure importance with a small amount of data and subsequently uses low-rank approximation (Hu et al., 2021) to partially recover lost knowledge after pruning. LoRAPrune (Zhang et al., 2023) combines parameter-efficient tuning methods with pruning to enhance performance on downstream tasks, which presents a pruning criterion based on LoRA, using the weights and gradients of LoRA for importance estimation instead of relying on pre-trained weights. Sheared LLaMA (Xia et al., 2023) also focused on structured pruning and introduces dynamic batch loading, utilizing losses across different domains to dynamically adjust the components of the sampled data in each training batch. LoRAShear (Chen et al., 2023b) proposed an innovative structure sparse optimizer called LoRA Half-Space Projected Gradient for progressive structured pruning and knowledge transfer. It employs a multi-stage knowledge recovery mechanism to effectively narrow down the performance gap between the full and compressed transformers.

**Semi-structured pruning.** Semi-structured pruning is an under-explored strategy that stays between unstructured pruning and structured pruning, exemplified by N:M sparsity (Zhou et al., 2020) that every contiguous set of M elements precisely contains N non-zero elements in a certain layer or weight matrix. SR-STE (Zhou et al., 2020) proposed the sparse-refined straight-through estimator to enhance the induction of

N:M sparsity in the network. A100 GPUs and the 2:4 fine-grained structured sparsity scheme, introduced by Nvidia (Choquette et al., 2021), which enable sparse neural networks to be accelerated on specific hardware. Semi-structured pattern retains relatively high model accuracy while facilitating efficient compression.

**Unstructured pruning.** Unstructured pruning centers on pruning less salient and individual parameters, wherever they are, offering greater flexibility compared to structured pruning. SparseGPT (Frantar & Alistarh, 2023) is an approach for few-shot pruning in transformers that avoids the need for retraining, being able to prune up to 60% of the parameters with minimal perplexity increase. It frames pruning as a sparse regression problem and addresses it using an approximate solver based on the inversion of the Hessian matrix. Syed et al. (2023) introduces an iterative pruning technique that fine-tunes the model during pruning with minimal training steps. Wanda (Sun et al., 2023) performs weight pruning through the product of weight magnitudes and their corresponding input activations. This method prunes the network with a single forward pass without depending on second-order information or requiring weight update, achieving competitive performance compared to SparseGPT. Shao et al. (2023) proposed Hessian sensitivity-aware mixed sparsity pruning to attain at least 50% sparsity in transformers without retraining.

## 3 Mixed Sparsity Training

This section introduces Mixed Sparsity Training (MST), our innovative approach for transformer pretraining. MST seamlessly integrates Dynamic Sparse Training (DST) with Sparsity Variation (SV) and Hybrid Sparse Attention (HSA). Throughout the training, the model evolves dynamically with a sparse topology. The sparsity variation is firstly elaborated in Section 3.1, and a novel topology evolution scheme, Mixed-Growing (MG), is given in Section 3.2. The hybrid sparse attention mechanism is detailed in Section 3.3. The nuanced integration of dynamic sparse training with sparsity variation and hybrid sparse attention positions MST as a self-consistent method, offering a potent means to optimize training processes and enhance the pretraining efficiency of transformers.

### 3.1 Sparsity Variation

This subsection provides a detailed explanation of the three phases in sparsity variation. The sparsity variation plays a pivotal role in dynamically allocating model's parameters during the pretraining process, as shown in Figure 3.

**Warm-Up Phase** In this phase, the dense model systematically transitions into a sparse configuration, ensuring an appropriate sparse initialization. Drawing inspiration from prior research (Zhu & Gupta, 2018; Liu et al., 2021a), we implement a progressive pruning strategy over a series of $N$ iterations, gradually inducing sparsity in the dense network. The sparsity level $S_t$ for each iteration is determined by the cubic decay function:

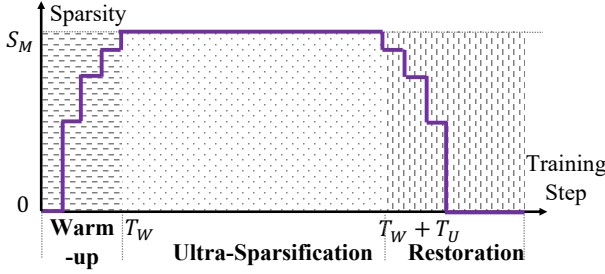

Figure 3: Sparsity variation.

$$S_t = S_M - S_M \cdot \left(1 - \frac{t}{N\Delta_W}\right)^3,$$

for $t \in [0, T_W]$, where $S_M$, $N$, and $\Delta_W$ represent the maximum sparsity, the number of sparsity stages, and the pruning frequency, respectively. The warm-up phase spans $T_W = N\Delta_W$ training steps. As mentioned by Zhu & Gupta (2018), the intuition behind this design is to quickly prune the network in the initial phase when redundant connections are abundant, and gradually reduce the number of weights pruned each time as fewer weights remain in the network, until the target sparsity is reached.

**Ultra-Sparsification Phase** The duration of this phase is $T_U$ training steps. We employ a novel topology evolution scheme termed Mixed-Growing (MG), detailed in Section 3.2, to facilitate proper sparse training.

Notably, the model is predominantly trained with a highly sparse topology during this phase, leveraging the observation that only a subset of parameters significantly contribute to model performance, while many parameters are redundant, as highlighted in prior works on transformer pruning (Ma et al., 2023; Frantar & Alistarh, 2023).

**Restoration Phase**  In this phase, connections are strategically reinstated to enhance the model's expressiveness and address potential performance loss incurred during former phases. This phase adopts a progressive approach symmetric to the warm-up phase, gradually reducing sparsity in the sparse network over a series of $N$ growing iterations. The sparsity for each iteration is determined by:

$$S_t = S_M \cdot \left(1 - \frac{t - T_W - T_U}{N\Delta_R}\right)^3,$$

for $t \in [T_W + T_U, T_W + T_U + N\Delta_R]$, where $\Delta_R$ denotes the growing frequency. The duration of the restoration phase is $T_R = N\Delta_R$ training steps. The progressive pruning and growing strategies have been empirically proven to achieve good performances in prior works (Zhu & Gupta, 2018; Liu et al., 2021a). Besides, our experiments in Section 4.2.1 also validates its superiority over other sparsity variation patterns.

*Remark* 3.1. The warm-up phase aims to establish a proper initial sparse topology, while the ultra-sparsification phase primarily focuses on reducing FLOPs. Subsequently, the restoration phase diligently addresses performance degradation resulting from sparsification. The interplay of three phases in our sparsity variation ensures a holistic transformation of the model, allowing it to leverage parameter redundancy while maintaining performance. The sparsity variation also forms the fundamental architecture of our proposed MST method.

### 3.2 Dynamic Sparse Training

We propose a novel topology evolution scheme called Mixed-Growing (MG), specifically designed for our MST methods, particularly effective in high-sparsity language models during the ultra-sparsification phase. We give a comparison of different DST schemes in Table 2, where existing methods exhibit weaknesses in transformers with massive amounts of parameters as shown in Section 4.1. By contrast, our proposed novel topology evolution scheme MG is dedicated to sparse training of transformers by introducing a more active exploration mechanism and outperforms other baselines.

Table 2: Comparison of different topology evolution schemes.

| Alg. | SET (Mocanu et al., 2018) | RigL (Evci et al., 2020) | MEST (Nowak et al., 2023) | **Mixed-Growing** (**Ours**) |
|---|---|---|---|---|
| Prune | $\min(|\theta|)$ | $\min(|\theta|)$ | $\min(|\theta| + \lambda|\nabla_\theta|)$ | $\min(|\theta|)$ |
| Grow | random | $\max(|\nabla_\theta|)$ | random | $(1-R)\max(|\nabla_\theta|)$ $+R \cdot \text{random}$ |

Our link growing policy utilizes a hybrid growth strategy that integrates gradient magnitude activation and random activation in a specific proportion to broaden the exploration range of weights. This strategy reduces the likelihood of the network to converge to sub-optimal structures. As for the link pruning, we adhere to a simple criterion based on weight magnitude in consideration of the findings by Nowak et al. (2023) regarding the minor differences among various pruning criteria in previous works. The weight magnitude pruning approach avoids the introduction of extra hyperparameters and works well in the pruning steps of our MG method.

Unlike existing topology evolution schemes, MG introduces flexibility by allowing a disparity between the number of pruned and grown links, facilitating adjustments in network sparsity based on predefined patterns like the one depicted in Figure 3. Specifically, MG utilizes current network sparsity $S_t$ and target

network sparsity $S'_t$ to determine the sparsity distribution for each layer, employing an Erdős–Rényi strategy (Mocanu et al., 2018). When the target sparsity $s'_l$ differs from the current sparsity $s_l$ for a given layer $l$, MG dynamically adjusts the number of growing or pruning links to match the target sparsity.

We exclude the training step index $t$ and provide the pseudocode of MG in Algorithm 1, where $M_\theta$ signifies the binary mask outlining the sparse network topology for $\theta$, $\zeta$ represents the topology update fraction, and $R$ denotes the ratio of connections activated through random growth. At each topology evolution step, MG eliminates a subset of existing connections with the smallest absolute weight values, while also prioritizing the revival of a certain number of empty connections with the largest gradients. The remaining connections are randomly generated for each layer.

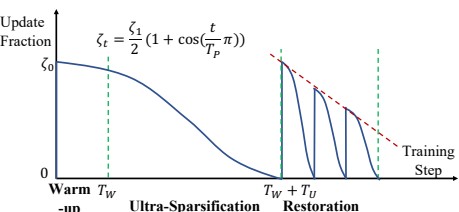

Figure 4: Piecewise cosine annealing.

The update fraction $\zeta$ is determined by a piecewise decaying cosine annealing scheme as

$$\zeta_t = \frac{\zeta_i}{2}\left(1 + \cos\left(\frac{t - T_{i-1}}{T_i - T_{i-1}}\pi\right)\right),$$

for $t \in [T_{i-1}, T_i)$, where $\zeta_i$ represents the update fraction magnitude, and $T_0 = 0, T_1 = T_W + T_U, T_2 = T_W + T_U + \Delta_R, \ldots, T_{N+1} = T_W + T_U + T_R$. This update schedule, visualized in Figure 4, ensures the network's robust evolutionary ability across different sparsity levels and is better than the single cosine annealing scheme employed in prior works (Dettmers & Zettlemoyer, 2019; Evci et al., 2020). An empirical comparison of different update schedules is provided in Appendix B.5.

---

**Algorithm 1** Topology Evolution by MG.
---
1: $\theta_l, N_l, s_l, s'_l$: parameters, number of parameters, current and target sparsity of layer $l$.
2: **for** each layer $l$ **do**
3:     $N_{\text{prune}} = \zeta(1 - s_l)N_l$
4:     $N_{\text{grow}} = N_{\text{prune}} + (s'_l - s_l)N_l$
5:     $N_{\text{rand}} = \lfloor N_{\text{grow}}R \rfloor$
6:     $N_{\text{grad}} = N_{\text{grow}} - N_{\text{rand}}$
7:     $\mathbb{I}_{\text{prune}} = \text{ArgTopK}(-|\theta_l \odot M_{\theta_l}|, N_{\text{prune}})$
8:     $\mathbb{I}_{\text{grad\_grow}} = \text{ArgTopK}_{i \notin \theta_l \odot M_{\theta_l} \setminus \mathbb{I}_{\text{prune}}}(|\nabla_{\theta_l}L|, N_{\text{grad}})$
9:     $\mathbb{I}_{\text{rand\_grow}} = \text{RandK}_{i \notin \theta_l \odot M_{\theta_l} \setminus (\mathbb{I}_{\text{drop}} \cup \mathbb{I}_{\text{grad\_grow}})}(N_{\text{rand}})$
10:    $\mathbb{I}_{\text{grow}} = \mathbb{I}_{\text{grad\_grow}} \cup \mathbb{I}_{\text{rand\_grow}}$
11:    Update $M_{\theta_l}$ according to $\mathbb{I}_{\text{prune}}$ and $\mathbb{I}_{\text{grow}}$
12:    $\theta_l \leftarrow \theta_l \odot M_{\theta_l}$
13: **end for**

---

### 3.3 Hybrid Sparse Attention

The majority of LLMs (Raffel et al., 2020; Brown et al., 2020; Du et al., 2021) are built upon transformer-based architectures, where the self-attention layer (Vaswani et al., 2017) plays a pivotal role in capturing spatial correlations within input texts. We first propose a novel unfactorized sparse strided self-attention, based on which we design the Hybrid Sparse Attention (HSA), to efficiently economize attention FLOPs while maintaining model performances.

A self-attention layer (Child et al., 2019) transforms a matrix of input embeddings $X$ into an output matrix, controlled by a connectivity pattern $S = \{S_1, \ldots, S_n\}$, where $n$ denotes the length of input embeddings $X$, and $S_i$ represents the set of indices to which the $i$-th output vector attends. Denote $W_q$, $W_k$, and $W_v$ as weight matrices responsible for transforming a given $x_i$ into a query, key, or value, respectively. The

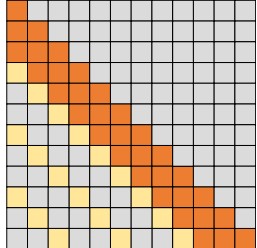

Figure 5: Strided attention with stride length $l = 3$.

self-attention output at each position is computed as the sum of values weighted by the scaled dot-product similarity of keys and queries:

$$\text{Attend}(X, S) = (a\,(x_i, S_i))_{i \in \{1, \ldots, n\}},$$

where $a(x_i, S_i) = \text{softmax}((W_q x_i)(W_k x_j)_{j \in S_i}^T / \sqrt{d}) \cdot (W_v x_j)_{j \in S_i}$, and $d$ represents the inner dimension. For autoregressive models such as GPT-2 (Brown et al., 2020), full self-attention sets $S_i = \{j : j \leq i\}$, allowing each element to attend to all previous positions, including its own.

Drawing inspiration from previous works on sparse self-attention (Child et al., 2019; Beltagy et al., 2020; Dai et al., 2019), we first introduce an *un-factorized sparse strided self-attention*, wherein different attention heads share the same sparse mask. While unfactorized sparse self-attention may entail more FLOPs than its factorized counterpart, it consistently maintains the same performance as dense self-attention, particularly evident in the ultra-sparsification phase in Section 4.2.3. The resultant FLOP savings of unfactorized sparse self-attention align with our targeted reduction of $4\times$, as demonstrated in Section 4.1. In our unfactorized strided sparse self-attention, each $i$-th output vector attends to the previous $l$ locations (depicted by orange cells

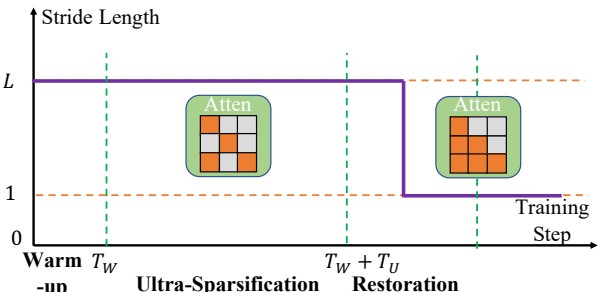

Figure 6: Stride length of attention mask in hybrid sparse attention. Notice that when the stride length equals 1, the model utilizes a dense mask.

in Figure 5), as well as every $l$-th location (illustrated by yellow cells in Figure 5), where $l$ denotes the chosen stride. Formally, $S_i = \{\max(i - l + 1, 0), \max(i - l + 2, 0), \ldots, i\} \cap \{j : (i - j) \bmod l = 0\}$ for $0 \leq i \leq n$. This pattern is visualized in Figure 5. Furthermore, our strided self-attention surpasses fixed self-attention patterns, as validated in Section 4.2.3, under the same FLOPs budget.

Based on the proposed unfactorized sparse self-attention, we are ready to propose the hybrid sparse attention to efficiently economize attention FLOPs while maintaining optimal model performance. In the initial stages of training, HSA employs an unfactorized strided sparse attention mask to conserve attention FLOPs. As the model becomes denser, HSA transitions to a dense attention mask, ensuring full restoration of model performance. Figure 6 visually illustrates the stride length of the attention mask in hybrid sparse attention during the training.

*Remark* 3.2. MST seamlessly integrates three interconnected components: the SV for temporal sparsification of MLP layers, MG for spatial sparsification of MLP layers, and HSA for optimizing self-attention layers. SV and MG work in tandem to harness the inherent *parameter redundancy* within transformers, dynamically shaping the model's connectivity both temporally and spatially throughout training. While SV orchestrates the temporal evolution of sparsity, MG fine-tunes spatial connectivity in real-time. In parallel, HSA addresses the *computational redundancy* inherent in self-attention mechanisms by distributing the operation across multiple steps, thereby minimizing FLOPs while preserving model performance. The concurrent operation of SV and MG ensures a comprehensive approach to sparsity management in MLP layers, with HSA enhancing computation efficiency in self-attention layers. This integrated framework enables the efficient and effective training of transformers, leveraging the synergies between temporal and spatial sparsification, and computational redundancy to achieve significant pretraining FLOP reductions without compromising model performance.

## 4 Experiment

This section outlines the experimental evaluation of our proposed MST, focusing on auto-regressive language modeling using GPT-2 (Brown et al., 2020), since GPT-2 and its variants are pivotal models in the domain of transformer-based LLMs. In addition, we also take experiments on BERT (Devlin et al., 2018) to validate the effectiveness of MST. The evaluation involves benchmarking MST's performance against baseline approaches on various language modeling tasks. We also conduct an ablation study of individual components in MST. The primary goal of this section is to showcase the advantages of MST in achieving a $4\times$ reduction in

pretraining FLOPs while maintaining performance comparable to dense models. The results are averaged on four random seeds and detailed experiment configurations are deferred to Appendix A. The code is available at `https://github.com/hupihe/Mixed-Sparsity-Training`.

## 4.1 Performance Comparison

Table 3: Performance comparison of different pretraining methods on GPT-2.

| Method | FLOPs | LAMBADA (ACC) | LAMBADA (PPL) | Wiki-2 (PPL) | PTB (PPL) | Wiki-103 (PPL) | 1BW (PPL) |
|---|---|---|---|---|---|---|---|
| Dense | 847.8G | 60.74 | 13.22 | 34.89 | 34.06 | 36.29 | 44.16 |
| Tiny | 212.7G | 53.29 | 31.91 | 58.00 | 58.71 | 60.78 | 71.96 |
| SS-80% | 267.7G | 50.59 | 54.77 | 83.98 | 75.98 | 88.00 | 99.44 |
| RigL-80% | 267.7G | 55.75 | 25.81 | 43.00 | 48.14 | 44.82 | 62.31 |
| **MST (Ours)** | 219.4G | 60.54 | 13.67 | 33.33 | 34.64 | 34.95 | 45.90 |

We conducted extensive experiments to assess the performance of our MST method against various baseline approaches across a range of zero-shot tasks from Brown et al. (2020) and few-shot tasks from GLUE (Wang et al., 2018). Our baseline models encompass a range of pretraining strategies, including standard dense pretraining, a compact dense model (Tiny), static sparse pretraining with 80% sparsity (SS-80%), and dynamic sparse pretraining with 80% sparsity facilitated by RigL (Evci et al., 2020) (RigL-80%). Detailed benchmark configurations and baseline setups can be found in Appendix A. The experiment results of zero-shot tasks are summarized in Table 3, where we also present the pretraining FLOPs of different models. The corresponding wall-clock training time is reported in Appendix B.1. The results of few-shot tasks are deferred in Appendix B.2.

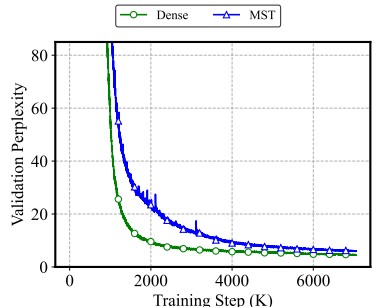

Figure 7: Perplexity of different pretraining methods on BERT.

Apart from GPT-2, we also conduct experiments on BERT (Devlin et al., 2018) to validate the effectiveness of MST. The training perplexity is shown in Figure 7, where we find that MST achieves performances comparable to dense training, with only less than one-third of the FLOPs required for dense training.

Under our setting, the tiny dense model and MST require only 25% pretraining FLOPs of the dense model, while the two sparse training methods (SS and RigL) require slightly more FLOPs, around 35%. However, among these four FLOPs-efficient training methods, only MST achieves performances comparable to those of the dense model. Moreover, MST outperforms the dense model in most zero-shot tasks. Notably, SS-80% is implemented based on SPDF (Thangarasa et al., 2023), while RigL-80% is built upon DynSparse (Dietrich et al., 2021). By the way, also note that the experimental results for the remaining three baselines (layer growth (Wang et al., 2023), block sparsification (Dao et al., 2022b), butterfly sparsification Dao et al. (2022c)) in Table 1 can be referenced from their respective papers, which are incomparable to our FLOP reduction ratio. The reason that we do not reproduce the experiment results from these three baselines is that they are beyond the scope of network sparsification. Besides, a comparison of our reproduced dense model with OpenAI's official GPT-2 checkpoint from HuggingFace is deferred to Appendix B.3.

## 4.2 Ablation Study

To gain insights into the contributions of each component in MST, we conducted an ablation study on sparsity variation, topology evolution schema, and hybrid sparse attention from the training pipeline.

### 4.2.1 Sparsity Variation

We evaluate the impact of different temporal sparsity variation on model performance during pretraining. Specifically, we compare several patterns: fully dense (Dense), fully sparse by MG (Sparse), sparse to dense (SD), dense to sparse to dense (DSD), gradual dense (GD), and our proposed Sparsity Variation (SV) utilizing a cubic decay function, as illustrated in Section 3.1. The validation perplexity, instant sparsity level, and average FLOP reduction compared to the dense model are plotted in Figure 8. As expected, the figure shows that fully dynamic sparse training with high sparsity throughout fails to match dense performance. However, when connections are grown back, all other four methods achieve dense-level performance in terms of validation perplexity. This underscores the importance of the restoration phase for sparse training methods of transformers to recover dense-comparable performance.

While GD demonstrates the fastest growth, it only achieves a 1.5× reduction in FLOPs. In contrast, SD achieves a higher FLOP reduction of over 2×. Moreover, the inclusion of an initial dense phase, as observed in DSD, outperforms SD after 120K in terms of validation perplexity with similar FLOP efficiency, underscoring the importance of sparse topology initialization for model performance. Our SV, employing a cubic decay function for sparsity variation, features a burn-in phase that is more efficient than sudden changes in sparsity, such as DSD, while maintaining comparable model performance. Thus, our proposed MST offers an efficient temporal sparsification pattern to achieve dense-comparable performance while simultaneously maximizing FLOP savings up to 2.7×. Additionally, we provide hyperparameter recommendations for SV in Appendix A.3.

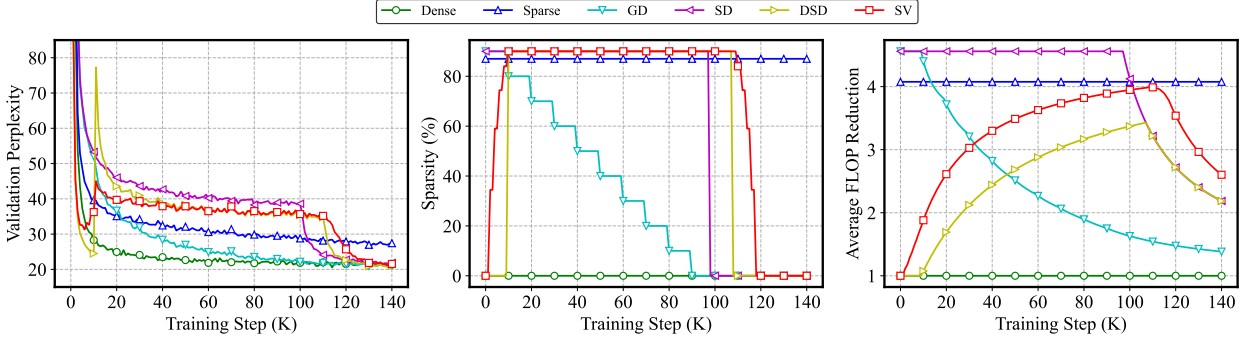

Figure 8: Ablation study on different sparsity variation patterns.

### 4.2.2 Topology Evolution Scheme

We showcase the effectiveness of our proposed topology evolution scheme, Mixed-Growing (MG), by comparing it with different topology evolution schemes in the task of training an ultra-sparse GPT-2 model with 90% sparsity. The baseline methods encompass static sparse training (SS), dynamic sparse training via SET (Mocanu et al., 2018), RigL (Evci et al., 2020), and MEST (Yuan et al., 2021), respectively.

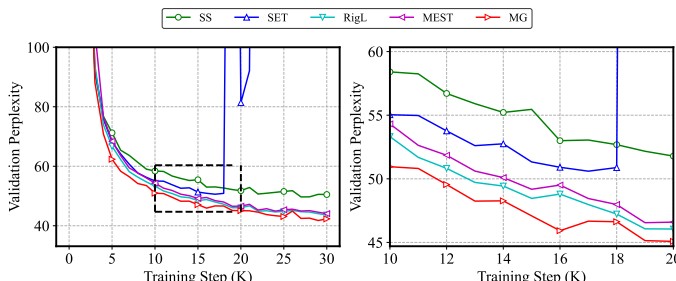

Figure 9: Perplexity of different topology evolution schemes.

Figure 9 provides an overview of the validation perplexity during the training of these methods, with the right subplot offering a closer examination of the highlighted region from the left subplot. SET consistently encounters loss spikes across all seeds occurring after 18K training steps, leading to outlier performance. Among the remaining methods, SS demonstrates the poorest performance, underscoring the critical importance of topology evolution during sparse training. RigL maintains its superiority over SET,

highlighting the importance of gradient-based growing strategies. However, it is worth noting that MEST underperforms RigL, which diverges from observations in smaller models (Nowak et al., 2023), suggesting that sparse training for transformers differs significantly from that of smaller models. Moreover, MG outperforms the other three methods, demonstrating its suitability for sparse training of transformers, attributed to its more proactive exploration mechanism, as elaborated in Section 4.3.

### 4.2.3 Sparse Self-Attention

We conducted a comparison of training performance among sparse models employing unfactorized strided self-attention and unfactorized fixed self-attention, as illustrated in Figure 10, with varying stride lengths. Specifically, we evaluated dense masks, fixed self-attention with stride lengths of 256 and 512, and strided self-attention with stride lengths of 128 and 256. Table 9 in Appendix A.5 presents the training FLOPs of the different models, while Figure 11 showcases their training performance.

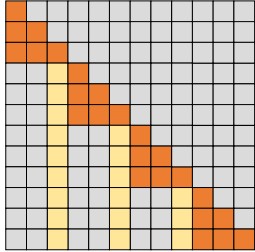

Figure 10: Fixed attention mask.

Notably, Fixed-256 and Strided-128 exhibit similar FLOPs, yet they both exhibit a performance gap compared to the dense mask. Conversely, Fixed-512 and Strided-256 demonstrate comparable performance to the dense mask. We opted for Strided-256 in our MST as it offers greater FLOP savings than Fixed-512 and achieves better performance than Strided-128.

### 4.3 Intuitive Insights

In addition to quantitative performance metrics, we present more intuitive insights into components of our proposed MST method. MST employs a sparsity variation comprising three phases: warm-up, ultra-sparsification, and restoration. The core concept of MST revolves around predominantly training the model during the ultra-sparsification phase to capitalize on the inherent parameter redundancy in transformers, thereby contributing significantly to FLOP reduction. The other two phases serve as auxiliary stages for supporting the ultra-sparsification phase to recover the full performance of dense training. The warm-up phase acts

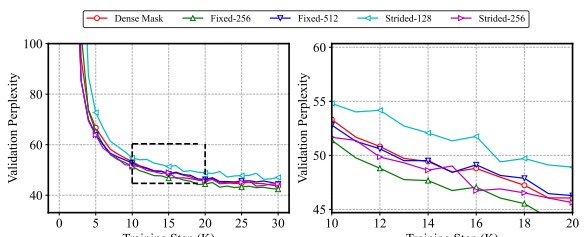

Figure 11: Validation perplexity of different attention patterns.

as a burn-in stage for the model to establish an optimal initial sparse topology, while the restoration phase endeavors to recover the performance loss incurred during the ultra-sparsification phase.

To underscore the necessity of mixed-growing in our ultra-sparsification phase, we initially present statistical results on parameter distribution across various scales of GPT-2 models. Subsequently, we visualize the evolution of model parameters during training to validate the effectiveness of our sparsity variation. These intuitive findings offer deeper insights into the operation of MST and its implications for efficient transformer pretraining.

**Why Mixed-Growing?** We present statistical results on parameter magnitude across various scales of GPT-2 models in Figure 12. We observe that the larger model exhibits a smaller standard deviation in parameter magnitudes. This suggests that transformers with a substantial number of parameters tend to have more homogeneous magnitudes. That is why we introduce a restoration phase to compensate for the performance

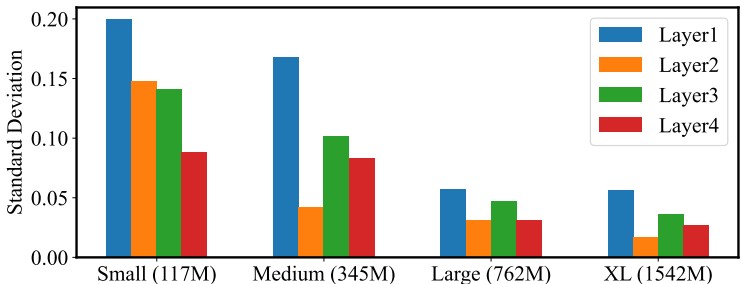

Figure 12: Standard deviation of parameters in the first encoder block in different scales of GPT-2 checkpoints from HuggingFace.

loss in the ultra-sparsification phase, as it may be harder to train a larger model with sparse neural networks throughout.

Moreover, the homogenization of parameters leads to reduced discrepancies in gradients, posing a potential challenge for gradient-based topology evolution schemes. With reduced differences in gradients, these schemes may become less suitable for guiding the evolution of sparse neural architectures. To counteract this limitation, we introduce the Mixed-Growing (MG) scheme, a tailored approach designed specifically for transformers. MG is crafted to inject additional random exploration into the evolution of sparse topology, ensuring its adaptability to the unique challenges presented by transformers. The efficacy of MG over alternative topology evolution schemes is demonstrated in Section 4.2.2.

**Why Sparsity Variation?** We visualize the model parameters through heatmaps after each phase in mixed sparsity training. For comparison, we include the visualization of a regularly trained dense model. Taking the weights of the projection matrix in the attention layers depicted in Figure 13 as an example, we observe horizontal bands in the parameters of the regularly trained dense model, indicating redundancies across output dimensions. Interestingly, the model trained using our mixed sparsity training exhibits similar bands in the same dimensions as the regularly trained dense model, albeit with narrower bandwidths. This suggests that our MST method effectively guides the model in learning essential parameters. Furthermore, the learned parameters are more concentrated, making them better suited for subsequent pruning in the post-training stage to facilitate faster inference of transformers, an advantage of our MST method.

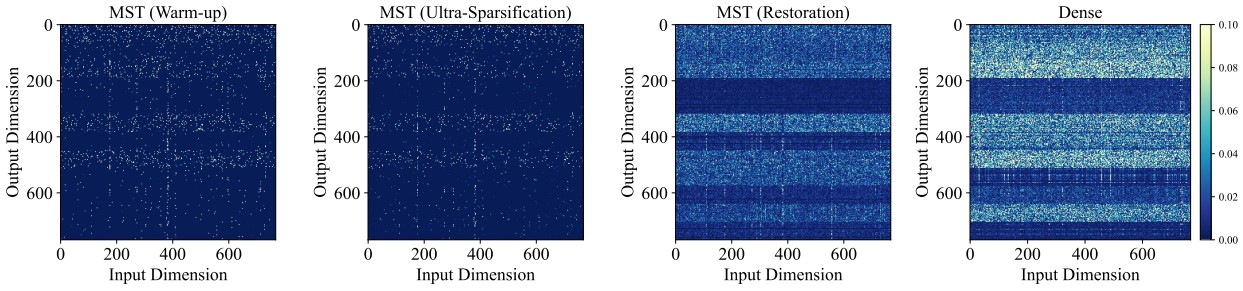

Figure 13: Heatmap of the weights of the projection matrix in the attention layer.

Additionally, the sparse topology obtained during the warm-up phase (first heatmap) displays aligned bands akin to the dense model, with minimal changes observed during the ultra-sparsification phase compared to the model after this phase (second heatmap). This underscores the effectiveness of the warm-up phase in establishing a favorable sparse initialization. However, we find that the restoration phase is also essential for restoring parameters, as the model parameters after the ultra-sparsification phase (second heatmap) still notably differ from those of the original dense model.

## 5 Conclusion and Future Work

Our proposed MST method presents a novel and effective approach for enhancing the efficiency of transformer pretraining. By seamlessly integrating Sparsity Variation (SV), Mixed-Growing (MG), and Hybrid Sparse Attention (HSA), MST offers a comprehensive solution to address both algorithmic inefficiencies and computational demands inherent in transformer pretraining. Through experimentation on GPT-2, we have demonstrated the efficacy of MST across various language modeling tasks, showcasing an exceptional FLOP reduction of 4× while maintaining the performance of dense models.

We envision that our discoveries lay the foundation for future research aimed at devising even more efficient and scalable approaches for training large-scale models. Extending the application of MST to larger scales or diverse model architectures holds substantial merit and can serve to corroborate its generalizability. Moreover, MST is an algorithmic and hardware-agnostic improvement that operates in neural connections

levels. Therefore, it is entirely orthogonal and can seamlessly integrate with existing system-level acceleration methods, such as training parallelism (Bian et al., 2021; Zhao et al., 2023), hardware-assisted attention operators (Dao et al., 2022c; Dao, 2023), and mixed precision training (Burgess et al., 2019; Liu et al., 2022), thus facilitating efficient transformer pretraining and achieving higher acceleration ratios.

## Acknowledgments

This work was support by the National Natural Science Foundation of China Grants 52450016 and 52494974.

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

## A  Experiment Details

### A.1  Hardware Setup

Our experiments are implemented with PyTorch 2.1.1 (Paszke et al., 2017), CUDA 12.1 (Kirk et al., 2007), run on 4× NVIDIA A100 Tensor Core GPUs. Each run needs about 24 hours to train the dense model for 100K steps.

### A.2  Model and Implementation Details

Considering OpenAI does not release the training dataset, WebText, of GPT-2 (Brown et al., 2020), we use the nanoGPT code base from https://github.com/karpathy/nanoGPT/. NanoGPT is a lightweight version of the GPT-2 model trained on the OpenWebText dataset (Gao et al., 2020). Our experiment implementation is derived from the small GPT-2 model. The architecture comprises 12 transformer layers and 12 attention heads, with an embedding size set to 768. The text is tokenized with the GPT-2 tokenizer (Brown et al., 2020). We adopt the train-validation split provided by nanoGPT. The training set comprises 9 billion tokens, and the validation set contains 4.4 million tokens. During training, we optimize the cross-entropy loss for next-token prediction. Consistent with nanoGPT, we employ GELU activations while disabling bias and Dropout. Distributed data parallelism with gradient accumulation is employed to enable a batch size of 480. Training is conducted with bfloat16 precision on machines with 4 A100 GPUs.

### A.3  Hyperparameter Settings of MST and baselines

We begin by detailing the hyperparameter settings for the dense model in Table 4, sourced from NanoGPT. These parameters are also utilized as the public parameters for the subsequent methods. Following this, we provide the private hyperparameter settings specific to MST, RigL, SS, and Tiny in Tables 5, 6, 7, and 8, respectively.

Table 4: Public hyperparameters.

| Hyperparameter | Value |
|---|---|
| Optimizer | AdamW |
| Activation function | GELU |
| Number of gradient accumulation steps | $5 \times 8$ |
| Batch size | 12 |
| Input sequence length | 1024 |
| Number of layers | 12 |
| Number of attention heads | 12 |
| Embedding dimensionality | 768 |
| Dropout rate | 0.0 |
| Bias | Not used |
| Learning rate | $6 \times 10^{-4}$ |
| Minimum learning rate | $6 \times 10^{-5}$ |
| Iteration intervals for learning rate decay | 140000 |
| Total number of training iterations | 140000 |
| Weight decay coefficient | 0.1 |
| Warmup steps | 2000 |
| Threshold value for gradient clipping | 0.1 |
| Exponential decay rate for the moving average of the gradient $\beta_1$ | 0.9 |
| Exponential decay rate for the moving average of the squared gradient $\beta_2$ | 0.95 |

Table 5: Private Hyperparameters of MST.

| Category | Hyperparameter | MST |
|---|---|---|
| SV | Maximum sparsity $S_M$ | 96% |
| | Number of sparsity levels N | 5 |
| | Pruning frequency $\Delta_W$ | 2000 |
| | Warm-up phase duration $T_W$ | 10000 |
| | Ultra-sparsification phase duration $T_U$ | 100000 |
| | Growing frequency $\Delta_R$ | 2000 |
| | Growing training phase duration $T_R$ | 10000 |
| DST | Initial topology update fraction $\zeta_1$ | 0.3 |
| | Topology update interval $\Delta_t$ | 100 |
| | Ratio of connections activated through random growth | 0.25 |
| HSA | Stride of attention mask | 256 |

Table 6: Private Hyperparameters of RigL.

| Category | Hyperparameter | RigL |
|---|---|---|
| DST | Sparsity $S_{\text{RigL}}$ | 80 |
| | Initial topology update fraction $\zeta_0$ | 0.3 |
| | Topology update interval $\Delta_t$ | 100 |

Table 7: Private Hyperparameters of SS.

| Category | Hyperparameter | SS |
|---|---|---|
| DST | Sparsity $S_{\text{SS}}$ | 80 |

Table 8: Private Hyperparameters of Tiny.

| Category | Hyperparameter | Tiny |
|---|---|---|
| Model Size | Embedding dimensionality | 384 |
| | Number of layers | 6 |

## A.4 Benchmark Details

In our performance evaluation, i.e. Table 12 in Section 4.1, we perform zero-shot evaluation of the models on 5 datasets and few-shot evaluation on 2 subtasks of GLUE. The detailed information about the datasets is elaborated in the following.

### A.4.1 Zero-Shot Tasks

**LAMBADA** The LAMBADA dataset (Paperno et al., 2016), sourced from BookCorpus, comprises 10,022 passages, which are further divided into 4,869 development passages and 5,153 test passages. It evaluates the capability of computational models to capture long-range dependencies and text comprehension. This task involves predicting the final word of sentences, which exhibits the feature wherein human participants can predict the final word when provided with the entire passage but struggle when given only the preceding sentence. To succeed on LAMBADA, computational models cannot simply rely on local context, but must demonstrate the capacity to track information from the broader discourse. The training dataset for language models evaluated on LAMBADA includes totalling 203 million words.

**PTB** The English Penn Treebank (PTB) corpus (Marcus et al., 1993) encompasses a diverse collection of English text drawn from sources such as news articles, magazines, and other publications, particularly the section corresponding to Wall Street Journal articles, which stands out as one of the most prominent and widely used datasets for evaluating sequence labelling models. This task involves annotating each word with its respective Part-of-Speech tag. In the conventional split of the corpus, sections 0 to 18 serve as the training set (comprising 38,219 sentences and 912,344 tokens), sections 19 to 21 function as the validation set (consisting of 5,527 sentences and 131,768 tokens), and sections 22 to 24 serve as the test set (comprising 5,462 sentences and 129,654 tokens).

**WikiText** The WikiText language modeling dataset (Merity et al., 2016) comprises a compilation of more than 100 million tokens extracted from a selection of Good and Featured articles on Wikipedia, commonly used in various language modeling tasks, including next-word prediction, text generation and text classification. Compared to the preprocessed version of Penn Treebank (PTB), WikiText-2 is more than twice the size, while WikiText-103 is over 110 times larger. Its composition of full articles makes it particularly suitable for models capable of capturing long-term dependencies.

**1BW** The One Billion Words (1BW) dataset (Chelba et al., 2013), originally introduced by the Google Brain team, is a substantial English language corpus for pretraining language models, which contains almost one billion words in the training data and is openly accessible for research purposes. This benchmark dataset covers various genres and topics ranging from news and technology to novels and is extensively used to assess the performance of statistical language models. Researchers leverage the 1BW dataset to pretrain language models, enhancing their performance across various downstream NLP tasks such as text classification, sentiment analysis, and language generation.

### A.4.2 Few-shot

General Language Understanding Evaluation (GLUE) (Wang et al., 2018) benchmark comprises a suite of diverse language understanding tasks, ranging from sentence-level to discourse-level, including tasks like text classification, sentence similarity, etc. It serves as a standardized platform for evaluating and comparing the performance of various NLP models, aiming to facilitate advancements in the field by providing a unified evaluation framework. Our experimental section uses 2 subtasks of the GLUE benchmark—RTE and MRPC.

**RTE** The Recognizing Textual Entailment (RTE) datasets come from a series of annual textual entailment challenges. The authors of the benchmark combined the data from RTE1 (Dagan et al., 2005), RTE2 (Haim et al., 2006), RTE3 (Giampiccolo et al., 2007), and RTE5 (Bentivogli et al., 2010). Examples are constructed based on news and Wikipedia text. The authors of the benchmark convert all datasets to a two-class split, where for three-class datasets they collapse neutral and contradiction into not entailment, for consistency. The RTE dataset is designed to assess models' understanding of textual entailment. The dataset sources texts from a series of annual textual entailment challenges. It consists of pairs of text, each comprising a premise and a hypothesis, where the task is to determine whether the premise logically entails the hypothesis.

**MRPC** The Microsoft Research Paraphrase Corpus (MRPC) (Dolan & Brockett, 2005) is a corpus of sentence pairs automatically extracted from online news sources. It is designed to evaluate models' performance in identifying semantic equivalence between pairs of sentences. The task involves determining whether the two sentences in each pair convey similar meanings.

### A.5 FLOP Calculation

As shown in Figure 1, the major FLOPs of the GPT-2 model come from fully connected layers and self-attention layers. We illustrate the computation of these two parts in the following. We disregard other layers which are effectively irrelevant or contribute minimally to the total computation overhead such as LayerNorm and Softmax. Furthermore, we exclude the FLOPs associated with the topology evolution process, as it occurs every 100 training steps, thus its influence on the final result is deemed negligible.

Initially, for a sparse network comprising $L$ fully connected layers and without considering bias terms, the required FLOPs for a forward pass can be computed as

$$\text{FLOPs}_{\text{forward}} = \sum_{l=1}^{L}(1 - S_l)(2I_l - 1)O_l \tag{1}$$

where $S_l$ is the sparsity, $I_l$ is the input dimensionality, and $O_l$ is the output dimensionality of the $l$-th layer. Note that Eq. (1) is also adopted in (Evci et al., 2020). When it comes to training FLOPs, the calculation involves multiple forward and backward passes across various networks. The FLOPs during the forward pass consist of two main components: the transformer and the final linear output layer, and the computation in the transformer primarily stems from the attention blocks and the multi-layer perceptron block.

Based on the model architecture, we can first delve into the details of the FLOP calculation of the attention mechanism blocks. We denote $L$ as the input sequence length, $N_{\text{embd}}$ as the embedding dimensionality, $D_{\text{head}}$ as the size of each attention head, $Q_{\text{atten}}$ as the ratio of FLOPs by sparse self-attention over the full self-attention, $S$ as the sparsity we pre-set. Besides, Table 9 shows the $Q_{\text{atten}}$ in different sparse self-attention patterns.

Table 9: FLOPs of different sparse self-attention patterns.

| Pattern | Dense Mask | Fixed 256 | Fixed 512 | Strided 128 | Strided 256 |
|---------|------------|-----------|-----------|-------------|-------------|
| $Q_{\text{atten}}$ (%) | 100.00 | 12.70 | 25.10 | 12.07 | 22.03 |

Thus, we have the following FLOPs:

1. Performing the projection to key, query, and value:

$$\text{FLOPs}_{\text{kqv}} = (1 - S) \times L \times ((2 \times N_{\text{embd}} - 1) \times 3 \times N_{\text{embd}})$$

2. Calculating the attention scores:

$$\text{FLOPs}_{\text{scores}} = Q_{\text{atten}} \times 2 \times L \times L \times N_{\text{embd}}$$

3. Aggregating the values through the reduction:

$$\text{FLOPs}_{\text{reduce}} = Q_{\text{atten}} \times 2 \times N_{\text{embd}} \times (L \times L \times D_{\text{head}})$$

4. Performing the final linear projection:

$$\text{FLOPs}_{\text{proj}} = (1 - S) \times L \times ((2 \times N_{\text{embd}} - 1) \times N_{\text{embd}})$$

Therefore, we can compute the total FLOPs for attention blocks as:

$$\text{FLOPs}_{\text{atten}} = \text{FLOPs}_{\text{kqv}} + \text{FLOPs}_{\text{scores}} + \text{FLOPs}_{\text{reduce}} + \text{FLOPs}_{\text{proj}}$$

Now, we shift focus to calculating the FLOPs for the MLP blocks. Before proceeding, we denote $D_{\text{ffw}}$ as the feed-forward size (the dimensionality of the hidden layer between the two linear layers in the MLP block), and it is commonly set to four times the value of $N_{\text{embd}}$. The calculation of FLOPs for the MLP involves summing the computational costs of two linear layers:

$$\text{FLOPs}_{\text{ffw1}} = (1 - S) \times L \times ((2 \times N_{\text{embd}} - 1) \times D_{\text{ffw}})$$
$$\text{FLOPs}_{\text{ffw2}} = (1 - S) \times L \times ((2 \times D_{\text{ffw}} - 1) \times N_{\text{embd}})$$

And we have

$$\text{FLOPs}_{\text{MLP}} = \text{FLOPs}_{\text{ffw1}} + \text{FLOPs}_{\text{ffw2}}$$

Therefore, the FLOPs of the transformer during the forward process is

$$\text{FLOPs}_{\text{transformer}} = N \times (\text{FLOPs}_{\text{atten}} + \text{FLOPs}_{\text{MLP}})$$

where $N$ denotes the number of layers (model depth), $\text{FLOPs}_{\text{atten}}$ and $\text{FLOPs}_{\text{MLP}}$ denote the FLOPs required in the attention and MLP blocks, respectively.

So far, we have elucidated the FLOP calculation for the internal modules of the transformer. During the forward pass, there is one remaining part, namely the final language model output layer, which is used to convert hidden representations into probability distributions over the vocabulary. We denote $D_{\text{vocal}}$ as the vocabulary size, and we have:

$$\text{FLOPs}_{\text{lm}} = (1 - S) \times L \times ((2 \times N_{\text{embd}} - 1) \times D_{\text{vocal}})$$

Then we obtain the total FLOPs for the forward process:

$$\text{FLOPs}_{\text{forward}} = \text{FLOPs}_{\text{transformer}} + \text{FLOPs}_{\text{lm}}$$

For the FLOPs of gradients backward propagation, denoted as $\text{FLOPs}_{\text{backward}}$, we compute it as twice the computational cost of the forward pass, which is adopted in existing literature (Evci et al., 2020), i.e., $\text{FLOPs}_{\text{backward}} = 2 \times \text{FLOPs}_{\text{forward}}$. Finally, the total FLOPs during the training process can be given by $\text{FLOPs}_{\text{total}} = \text{FLOPs}_{\text{forward}} + \text{FLOPs}_{\text{backward}}$.

# B  Supplementary Experiment Results

## B.1  Wall-Clock Training Time

Table 10: Wall-clock running time of different methods.

| Method | Running Time of 140K Training Steps |
|---|---|
| Dense | 30.10h |
| Tiny | 11.31h |
| SS-80% | 31.53h |
| RigL-80% | 32.63h |
| **MST (Ours)** | 34.32h |

## B.2  Performance comparison on Few-Shot Tasks

Table 11: Performance comparison on Few-Shot Tasks.

| Method | FLOPs | RTE (ACC) | MRPC (ACC) |
|---|---|---|---|
| Dense | 847.8G | 52.98 | 71.26 |
| Tiny | 212.7G | 55.87 | 71.63 |
| SS-80% | 267.7G | 50.00 | 68.50 |
| RigL-80% | 267.7G | 48.83 | 67.22 |
| **MST (Ours)** | 219.4G | 52.62 | 70.96 |

### B.3 Comparison with OpenAI's Checkpoint

A comparison between our reproduced dense model and NanoGPT, sourced from the official GPT-2 checkpoint by OpenAI via HuggingFace , is presented in Table 12. Upon review of Table 12, we observe that our model exhibits comparable performance to the OpenAI model across most tasks. However, in certain instances, such as on WikiText2 and WikiText103, our model demonstrates inferior performance. This discrepancy may be attributed to slight variations in model architecture and the utilization of different training datasets. While we employ the OpenWebText dataset, OpenAI utilizes the WebText dataset, which is not publicly available.

Table 12: Performance comparison of our reproduced GPT with with OpenAI's official GPT-2 checkpoint from HuggingFace.

| Method | LAMBADA (ACC) | LAMBADA (PPL) | WikiText2 (PPL) | PTB (PPL) | WikiText103 (PPL) | 1BW (PPL) |
|---|---|---|---|---|---|---|
| Dense (OpenAI) | 60.43 | 12.04 | 25.19 | 30.46 | 26.38 | 43.39 |
| Dense (Ours) | 60.74 | 13.22 | 34.89 | 34.06 | 36.29 | 44.16 |

### B.4 Training Curves of Comparative Evaluation in Section 4.1

The training curves of various methods evaluated in Section 4.1 are depicted in Figure 14. Upon examination of Figure 14, it becomes apparent that both Tiny and RigL fail to achieve performance comparable to the dense model. Additionally, the static sparse method encounters issues, evidenced by loss spikes across all four seeds. Notably, MST emerges as the sole method capable of reaching dense performance. Furthermore, the training curve of MST exhibits a spike at 100K training steps. This occurs due to the transition of the self-attention mask to a dense configuration at this moment within the hybrid sparse attention scheme. Despite this spike, MST rapidly converges back to a normal training trajectory autonomously, without requiring manual intervention for restoration.

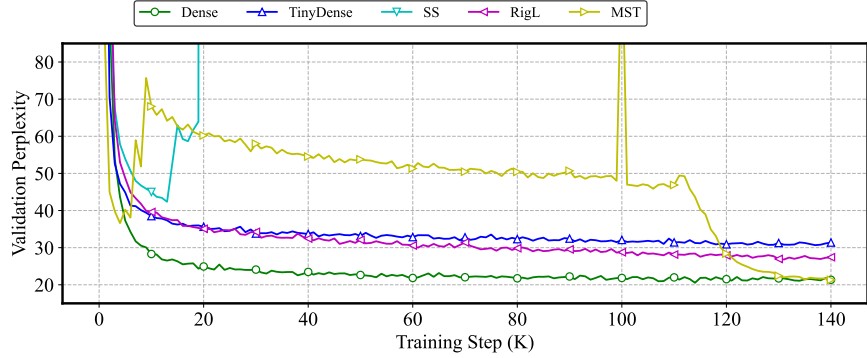

Figure 14: Training Curves of Comparative Evaluation in Section 4.1.

### B.5 Different Update Schedules in MST

We delved deeper into the impact of update schedules by training models with various schemes, including 1-cosine, 2-cosine, N-cosine, and Decay N-cosine. The experiment outcomes, depicted in Figure 15, reveal distinct performances across these schedules. The results indicate that the single cosine scheme struggles to maintain performance during the ultra-sparsification phase, while the 2-cosine scheme exhibits suboptimal performance during the restoration phase. Conversely, both N-cosine and Decay N-cosine schemes consistently yield good performance across both stages. This consistency can be attributed to the stepwise nature

of network sparsity in these stages, necessitating a reset of the update schedule to prevent convergence issues and subsequent performance degradation. Furthermore, the Decay N-cosine scheme outperforms the N-cosine scheme, as the decay of the update fraction helps stabilize the training process, thereby enhancing performance.

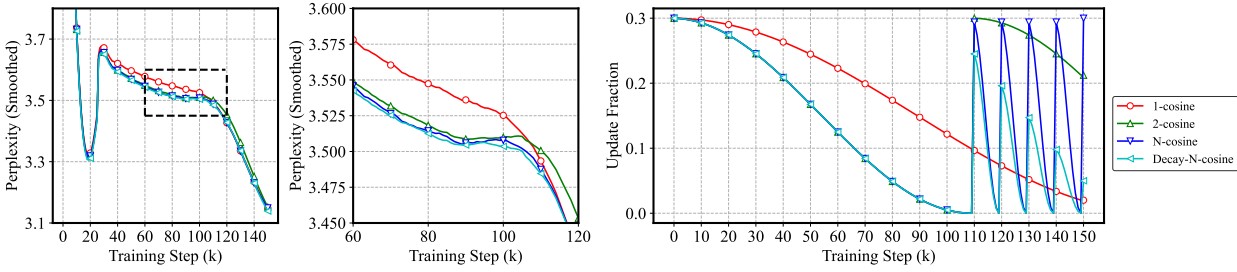

Figure 15: Ablation study on different mixed update schedules.

## B.6  Visualization of Model Parameters

We visualize the model parameters through heatmaps after each phase in mixed sparsity training. In addition to Figure 13 in Section 4.3, we present heatmaps of weights from various fully connected layers in Figures 16, 17, and 18. Figure 16 exhibits similar horizon bands as seen in Figure 13, indicating consistent patterns across layers. Conversely, parameter distributions in Figures 17 and 18 appear more uniform. Despite these differences, our model consistently learns parameters closely aligned with those of the dense model, underscoring the effectiveness of our MST method.

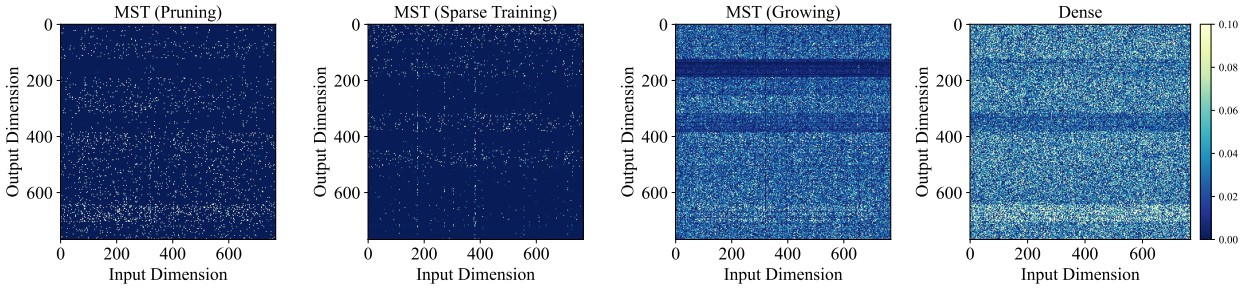

Figure 16: Heatmap of the weights of weight matrix in the second attention layer.

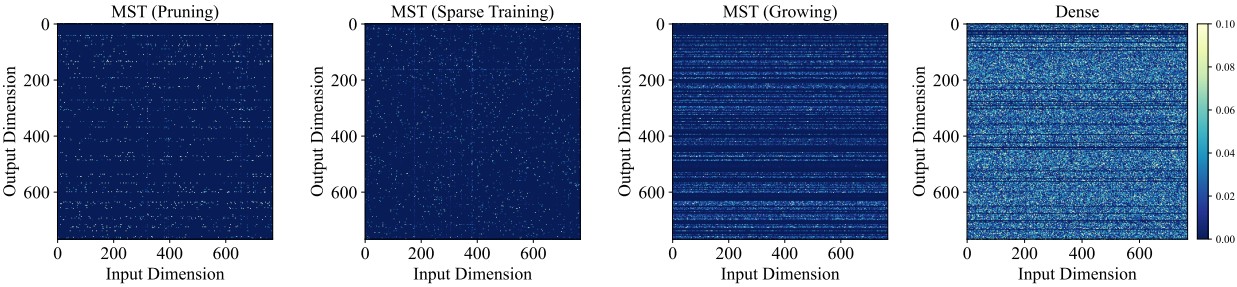

Figure 17: Heatmap of the weights of weight matrix in the first MLP layer.

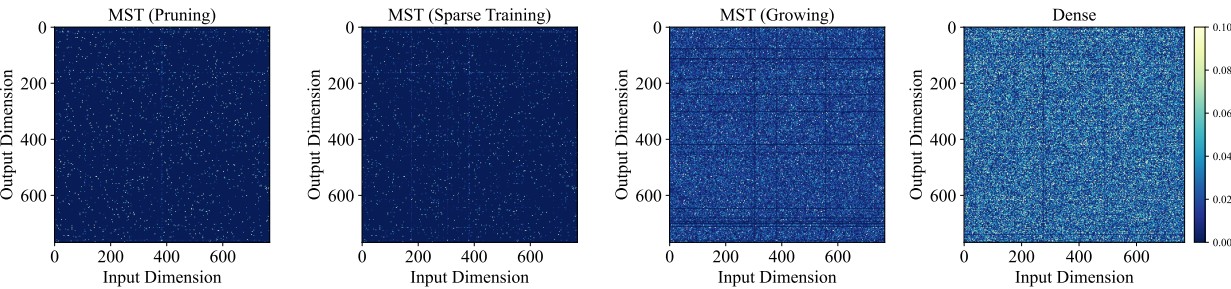

Figure 18: Heatmap of the weights of weight matrix in the second MLP layer.

