# OpenReview forum: "Mixed Sparsity Training: Achieving 4$\times$ FLOP Reduction for Transformer Pretraining"
_TMLR — Accepted by TMLR_

### Review · Reviewer_hWaJ · 2024-11-11

**Summary Of Contributions:**

This paper proposes a new pre-training method for LLMs called mixed sparsity training (MST), which not only maintains LLMs' performance but also minimizes training FLOPs. To better deal with the tradeoff between performance and training FLOPs, MST finds a way to integrate dynamic sparse training with Sparsity Variation and Hybrid Sparse Attention during the pretraining process and consists of three phases in the process, including warm-up, ultra-sparsification, and restoration. Specifically, the warm-up phase converts the dense neural network into a sparse one, the ultra-sparsification update model weights and sparsity topology with high sparsity and less training FLOPs, and the restoration phase reinstates connections for the final output neural network. Experiments demonstrate the benefits of the proposed method.

**Audience:**

Yes

**Claims And Evidence:**

Yes

**Requested Changes:**

* It would be good to add more explanation on the design of the sparsity determination formula for each iteration in the warm-up phase and restoration phase.
* It would be good to also add some real running time comparison, instead of the theoretical FLOPs comparison.
* It would be good to run more experiments on larger LLM models, like Llama.
* It would be good to include the standard deviation of the table values.
* It would be good if you could provide more discussion about why the proposed MST is orthogonal to training parallelism.
* The design of MST seems to also work for smaller DNNs. Could you provide some discussions about which part is tailored for LLM pre-training?
* You mentioned that there is a reduced discrepancy in gradients within the larger model. Could you provide more evidence for it?

**Strengths And Weaknesses:**

Strengths:
* The motivation is clear and the proposed method is well-defined.
* The problem tackled can contribute to the development of the LLM foundation model.
* Experiments show the improvement and demonstrate the benefit of the method.
* Some interesting insights are provided and discussed, which would be beneficial for the readers.

Weaknesses:
* The design of the sparsity determination formula for each iteration in the warm-up phase and restoration phase is not clear. Why do you choose the specific formula? Is the method sensitive to the formula design? Any ablation for this design?
* What are the additional FLOPs entailed from the unfactorized sparse self-attention compared to its factorized counterpart? How to balance the tradeoff between the two? I feel it to some extent similar to the relationship between structured and unstructured sparse training. But as we know, although unstructured sparse training theoretically also has good FLOPs reduction, it has strict requirements for the hardware, like GPU type. Does unfactorized sparse self-attention also have similar limitations?
* As mentioned in the previous comment, I was wondering if MST has a requirement for the hardware. Does it achieve real running time acceleration, instead of theoretical FLOPs?
* Experiments are only conducted on relatively small LLMs, like GPT-2.
* What is the sparsity rate for MST in Table 1? Would the trend be different when sparsity is further increased like 90% or reduced like 60%?
* You mention that the method is orthogonal to training parallelism. But I feel the ultra-sparsification phase as well as the model update during the phase would be influenced by the training parallelism.

---

> ### Author Response · Authors · 2025-01-21
> **Response to Reviewer hWaJ (Part 1)**
>
> Thanks for your time and effort in reviewing our paper! Please find our responses to your comments below. We will be happy to answer any further questions you may have.
>
> ---
>
> ### Weaknesses
>
> > **W1**:  The design of the sparsity determination formula for each iteration in the warm-up phase and restoration phase is not clear. Why do you choose the specific formula? Is the method sensitive to the formula design? Any ablation for this design?
>
> The cubic update formula used in the warm-up phase was proposed in [1] and has been adopted in subsequent works, such as [2]. The original formula presented in [1] is as follows:
> $$
>     s_t=s_f + (s_i-s_f)\left(1-\dfrac{t-t_0}{n\Delta t}\right)^3 \quad \text{for} \quad t\in\{t_0, t_0+\Delta t, ..., t_0 + n\Delta t\}
> $$
> The formula we use is a particular variant of the one above. Specifically, we set $t_0=0$, $s_i=0$, $s_f=S_M$, $\Delta t=\Delta_W$, and $n=N$. As mentioned in [1], the intuition behind this design is to quickly prune the network in the initial phase when redundant connections are abundant, and gradually reduce the number of weights pruned each time as fewer weights remain in the network, until the target sparsity is reached. The formula we use in restoration phase is symmetric to that in warm-up phase, which we believe helps maintain consistency between the model before and after sparsification.
>
> We conducted ablation experiments with different sparsity variation patterns in Figure 8 and found that this cubic pattern yielded the best results.
>
> In addition, this step is only performed to sparsify the model before ultra-sparsification phase, and after the main sparse training phase is completed, the model is restored to its dense form to ensure the performance. The main contribution of our work lies in proposing such dense-sparse-dense training paradigm, and during the sparse training phase, dynamically adjusting the network topology using the Mixed-Growing and Hybrid Sparse Attention method under the constraint of a given maximum sparsity level, in order to reduce training overhead. Therefore, we simply adopted a commonly used and effective approach for sparsifying the model before the sparse training phase. Whether there are better initial sparsification methods is also an interesting and open question, which we will explore in future work.
>
>
> ```
>  [1] Michael H Zhu and Suyog Gupta. To prune, or not to prune: Exploring the efficacy of pruning for model compression. 2018.
>  [2] Shiwei Liu, Tianlong Chen, Xiaohan Chen, Zahra Atashgahi, Lu Yin, Huanyu Kou, Li Shen, Mykola Pechenizkiy, Zhangyang Wang, and Decebal Constantin Mocanu. Sparse training via boosting pruning plasticity with neuroregeneration. Advances in Neural Information Processing Systems, 34:9908–9922, 2021a.
> ```
>
> > **W2**: What are the additional FLOPs entailed from the unfactorized sparse self-attention compared to its factorized counterpart? How to balance the tradeoff between the two? I feel it to some extent similar to the relationship between structured and unstructured sparse training. But as we know, although unstructured sparse training theoretically also has good FLOPs reduction, it has strict requirements for the hardware, like GPU type. Does unfactorized sparse self-attention also have similar limitations?
>
> Compared to its factorized counterpart, the additional FLOPs entailed from unfactorized sparse self-attention mainly come from computing the attention values for the previous $l$ consecutive positions. That is, it computes at most $l-1$ additional attention values compared to the factorized version. Therefore, by choosing an appropriate $l$, the trade-off between the two can be balanced. It is important to note that the purpose of unfactorized sparse self-attention is to maintain the model performance during sparse training through some reasonable dense computations (i.e., by computing the attention for the previous $l$ consecutive positions), while for positions beyond these $l$ consecutive ones, it remains consistent with the factorized attention. This behavior differs from the relationship between structured and unstructured sparse training. For this reason, unfactorized sparse self-attention should have few strict hardware requirements.

---

> > ### Author Response · Authors · 2025-01-21
> > **Response to Reviewer hWaJ (Part 2)**
> >
> > > **W3**: As mentioned in the previous comment, I was wondering if MST has a requirement for the hardware. Does it achieve real running time acceleration, instead of theoretical FLOPs?
> >
> > As sparse methods evolve alongside hardware co-design, the translation of FLOP reduction into wall-clock speedups becomes increasingly viable. Recent developments, such as specialized software kernels and dedicated hardware solutions like DeepSparse (NeuralMagic, 2021) and Cerebras CS-2 (Lie, 2022), signify promising strides towards realizing the benefits of unstructured sparsity during both training and inference stages (Thangarasa et al., 2023; Lasby et al., 2023).
> >
> > However, our primary objective is to demonstrate algorithmic advancements, and our findings suggest that training with MST can substantially reduce training FLOPs, stimulating further exploration towards physical implementations of sparse acceleration. Existing works also focus on algorithmic FLOP reduction, such as SparseGPT (Frantar & Alistarh, 2023) and Wanda (Sun et al., 2023). Our work, alongside these contributions, paves the way for more efficient and scalable LLMs. Thus, the translation of FLOP reduction into wall-clock speedups is beyond the scope of this paper.
> >
> > ```
> > (NeuralMagic, 2021) NeuralMagic. Deepsparse, 2021. URL https://github.com/neuralmagic/deepsparse.
> > (Lie, 2022) Lie, S. Harnessing the Power of Sparsity for Large GPT AI Models. https://www.cerebras.net/blog/harnessing-the-power-of-sparsity-forlarge-gpt-ai-models, 2022.
> > (Thangarasa et al., 2023) Thangarasa, V., Gupta, A., Marshall, W., Li, T., Leong, K., DeCoste, D., Lie, S., and Saxena, S. SPDF: Sparse pretraining and dense fine-tuning for large language models. In UAI, 2023.
> > (Lasby et al., 2023) Lasby, M., Golubeva, A., Evci, U., Nica, M., and Ioannou, Y. Dynamic sparse training with structured sparsity. arXiv, 2023.
> > ```
> >
> > > **W4**: Experiments are only conducted on relatively small LLMs, like GPT-2.
> >
> > We focus on GPT-2, a pivotal model in the domain of LLMs. Notably, existing research on LLM pruning, such as that by Frantar & Alistarh (2023) and Sun et al. (2023), also conducts experiments primarily on GPT-2. Our experiments with MST on GPT-2 showcase a remarkable FLOP reduction of $4\times$ without compromising performance across various zero-shot and few-shot downstream tasks. We firmly believe in the potential for MST to extend its efficacy to larger models with diverse architectures.
> >
> > Due to our current computing resource limitations, conducting experiments on pretraining processes for models (>7B) presents a challenge. The constraints we face also underscore the critical need to reduce the computational resources required by LLMs, which aligns with the core objective of our work. While we recognize that our current efforts may not fully achieve the ultimate goal of efficiently pretraining LLMs, we firmly believe that our work represents a pioneering step in this direction and holds meaningful implications for the research community.
> >
> > > **W5**: What is the sparsity rate for MST in Table 1? Would the trend be different when sparsity is further increased like 90% or reduced like 60%?
> >
> > As shown in Table 5 of the appendix, the sparsity used by MST during the Ultra-sparsification phase is 96%, which is quite a high level of sparsity. For other baselines, as the sparsity increases, the required FLOPs decrease somewhat, but the performance also drops. On the other hand, when sparsity decreases, the performance may approach that of the dense model, but the required FLOPs increase. However, MST achieves a balance by reducing FLOPs while maintaining performance close to that of the dense model. Ablation experiments for other sparsity levels will be included in the revised version.
> >
> > > **W6**: You mention that the method is orthogonal to training parallelism. But I feel the ultra-sparsification phase as well as the model update during the phase would be influenced by the training parallelism.
> >
> > MST is an algorithmic improvement that is hardware-agnostic, making it orthogonal to parallel training. For instance, our experiments were also conducted with parallel training on multiple GPUs.

---

> > > ### Author Response · Authors · 2025-01-21
> > > **Response to Reviewer hWaJ (Part 3)**
> > >
> > > ### Requested Changes
> > >
> > > > **RC1**: It would be good to add more explanation on the design of the sparsity determination formula for each iteration in the warm-up phase and restoration phase.
> > >
> > > We will provide a more detailed explanation of the reasoning behind this design in our revision.
> > >
> > > > **RC2**: It would be good to also add some real running time comparison, instead of the theoretical FLOPs comparison.
> > >
> > > We will include some real running time comparison in our revision.
> > >
> > > > **RC3**: It would be good to run more experiments on larger LLM models, like Llama.
> > >
> > > We will include more experiments on larger LLM models in our revision.
> > >
> > > > **RC4**: It would be good to include the standard deviation of the table values.
> > >
> > > We will include the standard deviation in our revision.
> > >
> > > > **RC5**: It would be good if you could provide more discussion about why the proposed MST is orthogonal to training parallelism.
> > >
> > > We will provide a more more discussion about this in our revision.
> > >
> > > > **RC6**: The design of MST seems to also work for smaller DNNs. Could you provide some discussions about which part is tailored for LLM pre-training?
> > >
> > > Existing sparsification methods fix the sparsity level, which makes them effective for small DNNs but can lead to performance degradation in LLMs due to parameter pruning. To tackle the unique challenges associated with pre-training LLMs, our MST design introduces a dense-sparse-dense training paradigm, effectively reducing training overhead while preserving the final performance of LLMs. The application of MST to small DNNs is also an intriguing direction, which we leave for future work.
> > >
> > > > **RC7**: You mentioned that there is a reduced discrepancy in gradients within the larger model. Could you provide more evidence for it?
> > >
> > > We present statistical results on parameter magnitudes across various scales of GPT-2 models in Figure 12. We observe that the larger model exhibits a smaller standard deviation in parameter magnitudes. This suggests that there is a greater number of homogeneous parameters, which leads to reduced discrepancies in gradients.
> > >
> > > ---
> > > We are grateful for your constructive suggestions, which have significantly guided our improvements. We hope our response addresses your concerns. We will also be happy to answer any further questions you may have. Thank you very much!

---

### Review · Reviewer_Uasj · 2025-01-07

**Summary Of Contributions:**

This paper introduces Mixed Sparsity Training (MST), a novel approach to significantly reduce the computational cost of pretraining large transformer-based models like GPT-2 while maintaining performance. MST integrates three techniques—Dynamic Sparse Training (DST), Sparsity Variation (SV), and Hybrid Sparse Attention (HSA)—to achieve a 4× reduction in FLOPs. The training process is divided into three phases: warm-up, which transitions the dense model to a sparse topology; ultra-sparsification, where the model is predominantly trained in a sparse configuration; and restoration, which recovers potential performance loss by reinstating connections. By dynamically adjusting sparse connections and using a sparse attention mask that transitions to dense over time, MST addresses algorithmic inefficiencies inherent in pretraining.

**Audience:**

Yes

**Broader Impact Concerns:**

There is no outstanding ethical issues from this paper.

**Claims And Evidence:**

Yes

**Requested Changes:**

- Consider including wall-clock time comparisons and analyses in the experiments to provide a clearer understanding of the practical speedup.
- Add a discussion on hardware implementation feasibility and its implications to better contextualize the method's real-world applicability.
- Expand the experiments to include models beyond BERT and GPT-2 to demonstrate the broader applicability of the proposed method.

**Strengths And Weaknesses:**

Strengths:

- The proposed method achieves a 4× reduction in computational cost during pretraining without compromising model performance, a significant saving given the increasing computational demands of AI models.
- Dynamic Sparse Training (DST), Sparsity Variation (SV), and Hybrid Sparse Attention (HSA) are presented as innovative and promising techniques.
- The method is compatible with other system-level optimizations, such as mixed-precision training and distributed parallelism, enabling additional efficiency gains.


Weaknesses:

- While the method effectively demonstrates a reduction in FLOPs, it remains unclear if this translates into tangible end-to-end speedups. Including wall clock time comparisons would strengthen the analysis.
- The efficiency of implementing the proposed scheme on hardware is uncertain. Although the authors suggest seamless integration with methods like flash attention, the extent of actual speedup achievable on hardware is not adequately demonstrated.

---

> ### Author Response · Authors · 2025-01-21
> **Response to Reviewer Uasj**
>
> Thanks for your time and effort in reviewing our paper! Please find our responses to your comments below. We will be happy to answer any further questions you may have.
>
> ---
>
> ### Weaknesses
>
> > **W1**: While the method effectively demonstrates a reduction in FLOPs, it remains unclear if this translates into tangible end-to-end speedups. Including wall clock time comparisons would strengthen the analysis.
>
> As sparse methods evolve alongside hardware co-design, the translation of FLOP reduction into wall-clock speedups becomes increasingly viable. Recent developments, such as specialized software kernels and dedicated hardware solutions like DeepSparse (NeuralMagic, 2021) and Cerebras CS-2 (Lie, 2022), signify promising strides towards realizing the benefits of unstructured sparsity during both training and inference stages (Thangarasa et al., 2023; Lasby et al., 2023).
>
> However, our primary objective is to demonstrate algorithmic advancements, and our findings suggest that training with MST can substantially reduce training FLOPs, stimulating further exploration towards physical implementations of sparse acceleration. Existing works also focus on algorithmic FLOP reduction, such as SparseGPT (Frantar & Alistarh, 2023) and Wanda (Sun et al., 2023). Our work, alongside these contributions, paves the way for more efficient and scalable LLMs. Thus, the translation of FLOP reduction into wall-clock speedups is beyond the scope of this paper.
>
> ```
> (NeuralMagic, 2021) NeuralMagic. Deepsparse, 2021. URL https://github.com/neuralmagic/deepsparse.
> (Lie, 2022) Lie, S. Harnessing the Power of Sparsity for Large GPT AI Models. https://www.cerebras.net/blog/harnessing-the-power-of-sparsity-forlarge-gpt-ai-models, 2022.
> (Thangarasa et al., 2023) Thangarasa, V., Gupta, A., Marshall, W., Li, T., Leong, K., DeCoste, D., Lie, S., and Saxena, S. SPDF: Sparse pretraining and dense fine-tuning for large language models. In UAI, 2023.
> (Lasby et al., 2023) Lasby, M., Golubeva, A., Evci, U., Nica, M., and Ioannou, Y. Dynamic sparse training with structured sparsity. arXiv, 2023.
> ```
>
> > **W2**: The efficiency of implementing the proposed scheme on hardware is uncertain. Although the authors suggest seamless integration with methods like flash attention, the extent of actual speedup achievable on hardware is not adequately demonstrated.
>
> Since our MST operates in neural connections levels, it is entirely orthogonal and can seamlessly integrate with existing system-level acceleration methods, such as training parallelism (Zhao et al., 2023), hardware-assisted attention operators (Dao, 2023), and mixed precision training (Liu et al., 2022), thus facilitating efficient transformer pretraining and achieving higher acceleration ratios.
>
> ### Requested Changes
>
> > **RC1**: Consider including wall-clock time comparisons and analyses in the experiments to provide a clearer understanding of the practical speedup.
>
> We will include wall-clock time comparisons and analyses in the experiments in our revision.
>
> > **RC2**: Add a discussion on hardware implementation feasibility and its implications to better contextualize the method's real-world applicability.
>
> We will include a discussion on hardware implementation feasibility and its implications in our revision.
>
> > **RC3**: Expand the experiments to include models beyond BERT and GPT-2 to demonstrate the broader applicability of the proposed method.
>
> We will expand the experiments to include models in our revision.
>
> ---
> We are grateful for your constructive suggestions, which have significantly guided our improvements. We hope our response addresses your concerns. We will also be happy to answer any further questions you may have. Thank you very much!

---

### Review · Reviewer_HPPd · 2025-01-07

**Summary Of Contributions:**

The paper introduces Mixed Sparsity Training (MST), a method for reducing the computational overhead of pretraining transformers by dynamically integrating sparsity techniques. The proposed method, MST, leverages Sparsity Variation (SV), Mixed-Growing (MG), and Hybrid Sparse Attention (HSA) through three phases: warm-up, ultra-sparsification, and restoration. It achieves a 4x reduction in FLOPs without compromising performance, as demonstrated using GPT-2 across various tasks. MST is compatible with existing acceleration techniques such as training parallelism, mixed precision, and hardware-assisted methods.


The key claimed contributions of this paper are:
1. proposes a Mixed Sparsity Training (MST) framework combining multiple dynamic sparsity techniques.
2. demonstrates up to a 4x reduction in FLOPs while retaining comparable performance to dense models.
3. experiment to validates MST through experiments on GPT-2, showing effectiveness across zero-shot and few-shot tasks.

**Audience:**

Yes

**Broader Impact Concerns:**

None.

**Claims And Evidence:**

No

**Requested Changes:**

See the **Weaknesses**.

**Strengths And Weaknesses:**

**Strengths**
1. Theoretically, the proposed method MST achieves a 4x FLOP reduction
2. Experiments highlight the effectiveness of Sparsity Variation (SV), Mixed-Growing (MG), and Hybrid Sparse Attention (HSA) through comprehensive ablations
3. This paper claim that MST is compatible with other acceleration techniques, such as training parallelism, hardware-assisted attention operators, and mixed precision training, enabling broader adoption.










**Weaknesses**
1. Dynamic Sparse Training (DST) is challenging to implement in real-world scenarios as memory usage decreases gradually during training, requiring additional engineering efforts to handle memory optimizations effectively. More explanation needed for this point.
2. The claimed FLOP reductions may be theoretical and difficult to achieve in practical settings without experimental validation of real training speed improvements. Adding training time as the x-axis in Figures 7, 8, 9, and 11 could provide better evidence of the effectiveness of the proposed method.
3. This paper only presents the experimental results on GPT-2; its performance on larger or other types of LLMs, like Llama, is not explored. It would be better if experiments on more LLM architectures could be conducted.
4. This paper claims that MST is orthogonal to other system-level accelerations, such as training parallelism, hardware-assisted attention operators, and mixed precision training. I am interested in this point. For example, how can Hybrid Sparse Attention be combined with FlashAttention to increase FLOPs/s? Claims of MST’s compatibility with other accelerations, such as training parallelism and hardware-assisted methods, lack demonstration. Specific evaluations, such as integrating Hybrid Sparse Attention with FlashAttention, could strengthen this claim.
5. Not strong support for “transformers exhibit considerable redundancy in pretraining computations”. It would be better to provide have empirical support for the claim that transformers exhibit redundancy in pretraining computations.
6.  The proposed method MST primarily combines existing methods, such as Dynamic Sparse Training and Hybrid Sparse Attention, which may limit its perceived novelty.

---

> ### Author Response · Authors · 2025-01-21
> **Response to Reviewer HPPd (Part 1)**
>
> Thanks for your time and effort in reviewing our paper! Please find our responses to your comments below. We will be happy to answer any further questions you may have.
>
> ---
>
> ### Weaknesses
>
> > **W1**: Dynamic Sparse Training (DST) is challenging to implement in real-world scenarios as memory usage decreases gradually during training, requiring additional engineering efforts to handle memory optimizations effectively. More explanation needed for this point.
>
> As sparse methods evolve alongside hardware co-design, the translation of FLOP reduction into wall-clock speedups becomes increasingly viable. Recent developments, such as specialized software kernels and dedicated hardware solutions like DeepSparse (NeuralMagic, 2021) and Cerebras CS-2 (Lie, 2022), signify promising strides towards realizing the benefits of unstructured sparsity during both training and inference stages (Thangarasa et al., 2023; Lasby et al., 2023).
>
> However, our primary objective is to demonstrate algorithmic advancements, and our findings suggest that training with MST can substantially reduce training FLOPs, stimulating further exploration towards physical implementations of sparse acceleration. Existing works also focus on algorithmic FLOP reduction, such as SparseGPT (Frantar & Alistarh, 2023) and Wanda (Sun et al., 2023). Our work, alongside these contributions, paves the way for more efficient and scalable LLMs. Thus, the translation of FLOP reduction into wall-clock speedups is beyond the scope of this paper.
>
> ```
> (NeuralMagic, 2021) NeuralMagic. Deepsparse, 2021. URL https://github.com/neuralmagic/deepsparse.
> (Lie, 2022) Lie, S. Harnessing the Power of Sparsity for Large GPT AI Models. https://www.cerebras.net/blog/harnessing-the-power-of-sparsity-forlarge-gpt-ai-models, 2022.
> (Thangarasa et al., 2023) Thangarasa, V., Gupta, A., Marshall, W., Li, T., Leong, K., DeCoste, D., Lie, S., and Saxena, S. SPDF: Sparse pretraining and dense fine-tuning for large language models. In UAI, 2023.
> (Lasby et al., 2023) Lasby, M., Golubeva, A., Evci, U., Nica, M., and Ioannou, Y. Dynamic sparse training with structured sparsity. arXiv, 2023.
> ```
>
> > **W2**: The claimed FLOP reductions may be theoretical and difficult to achieve in practical settings without experimental validation of real training speed improvements. Adding training time as the x-axis in Figures 7, 8, 9, and 11 could provide better evidence of the effectiveness of the proposed method.
>
> We will include training time in the experiment in our revision.
>
> > **W3**: This paper only presents the experimental results on GPT-2; its performance on larger or other types of LLMs, like Llama, is not explored. It would be better if experiments on more LLM architectures could be conducted.
>
> We focus on GPT-2, a pivotal model in the domain of LLMs. Notably, existing research on LLM pruning, such as that by Frantar & Alistarh (2023) and Sun et al. (2023), also conducts experiments primarily on GPT-2. Our experiments with MST on GPT-2 showcase a remarkable FLOP reduction of $4\times$ without compromising performance across various zero-shot and few-shot downstream tasks. We firmly believe in the potential for MST to extend its efficacy to larger models with diverse architectures.
>
> Due to our current computing resource limitations, conducting experiments on pretraining processes for models (>7B) presents a challenge. The constraints we face also underscore the critical need to reduce the computational resources required by LLMs, which aligns with the core objective of our work. While we recognize that our current efforts may not fully achieve the ultimate goal of efficiently pretraining LLMs, we firmly believe that our work represents a pioneering step in this direction and holds meaningful implications for the research community.
>
> > **W4**: This paper claims that MST is orthogonal to other system-level accelerations, such as training parallelism, hardware-assisted attention operators, and mixed precision training. I am interested in this point. For example, how can Hybrid Sparse Attention be combined with FlashAttention to increase FLOPs/s? Claims of MST’s compatibility with other accelerations, such as training parallelism and hardware-assisted methods, lack demonstration. Specific evaluations, such as integrating Hybrid Sparse Attention with FlashAttention, could strengthen this claim.
>
> MST is an algorithmic improvement that is hardware-agnostic, making it orthogonal to system-level accelerations. However, our experiments were also conducted with mixed precision training and parallel training on multiple GPUs, which precisely demonstrates the compatibility of MST with system-level acceleration methods.

---

> > ### Author Response · Authors · 2025-01-21
> > **Response to Reviewer HPPd (Part 2)**
> >
> > > **W5**: Not strong support for “transformers exhibit considerable redundancy in pretraining computations”. It would be better to provide have empirical support for the claim that transformers exhibit redundancy in pretraining computations.
> >
> > Through experiments, we demonstrate that MST can achieve comparable performance to dense models with significantly smaller computational overhead, which precisely proves the existence of substantial redundancy in transformers. Specifically, MST addresses redundancy in the fully connected layers through Sparsity Variation and Dynamic Sparse Training, and reduces redundancy in the attention mechanism through Hybrid Sparse Attention.
> >
> > > **W6**: The proposed method MST primarily combines existing methods, such as Dynamic Sparse Training and Hybrid Sparse Attention, which may limit its perceived novelty.
> >
> > Our paper introduces a pioneering pretraining technique named MST, which integrates dynamic sparse training and hybrid sparse attention throughout the pretraining process. The contribution lies in proposing a robust method to achieve efficient training for transformer-based LLMs. We also propose a novel topology evolution scheme called Mixed-Growing (MG), specifically designed for our MST methods.
> >
> > ---
> > We are grateful for your constructive suggestions, which have significantly guided our improvements. We hope our response addresses your concerns. We will also be happy to answer any further questions you may have. Thank you very much!

---

> > > ### Comment · Reviewer_HPPd · 2025-01-22
> > > **Thank you for your response to my review.**
> > >
> > > Dear Authors,
> > >
> > > I greatly appreciate the authors' response to my review. After carefully reading your response and other reviews, I have the following concerns:
> > >
> > > 1. **For W1**, I am not convinced by the statement:
> > >    *"Our work paves the way for more efficient and scalable LLMs. Thus, the translation of FLOP reduction into wall-clock speedups is beyond the scope of this paper."*
> > >    Since the main claim of the proposed method is its "efficiency," evidence of wall-clock speedup is essential.
> > >
> > > 2. **For W2**, you mentioned,
> > >    *"We will include training time in the experiment in our revision."*
> > >    Could you please upload your revision or share the results in the review thread here?
> > >
> > > 3. **For W4**, my question,
> > >    *"how can Hybrid Sparse Attention be combined with FlashAttention to increase FLOPs/s? Claims of MST’s compatibility with other accelerations, such as training parallelism and hardware-assisted methods, lack demonstration. Specific evaluations, such as integrating Hybrid Sparse Attention with FlashAttention, could strengthen this claim"*
> > >    has not been answered.
> > >
> > > 4. **For W3**, while I understand your resource limitations, experimenting with a 1B LLaMA or a 3B model seems feasible and would strengthen your work.
> > >
> > > I am mostly satisfied with your responses to the other weaknesses. Thank you again for addressing my review.
> > >
> > >
> > > Reviewer HPPd

---

> > > > ### Author Response · Authors · 2025-02-04
> > > > **Response to the Reply of Reviewer HPPd**
> > > >
> > > > Thanks for your time and effort in replying our response!
> > > >
> > > > ---
> > > >
> > > > ### R1
> > > >
> > > > > **For W1**, I am not convinced by the statement:
> > > > *"Our work paves the way for more efficient and scalable LLMs. Thus, the translation of FLOP reduction into wall-clock speedups is beyond the scope of this paper."*
> > > > Since the main claim of the proposed method is its "efficiency," evidence of wall-clock speedup is essential.
> > > >
> > > > We recognize that wall-clock time acceleration is crucial for training "efficiency", but optimizations at the algorithmic level and corresponding hardware support do not always develop in sync. Therefore, for existing *efficient* unstructured sparsification methods, such as SparseGPT (Frantar & Alistarh, 2023) and Wanda (Sun et al., 2023), it is difficult to directly translate FLOPS acceleration for sparse training into wall-clock time acceleration. This is why many works, including ours, only report FLOPS acceleration. However, our work at the algorithmic level provides a new optimization approach, which we believe can inspire more future work at the hardware level.
> > > >
> > > > ```
> > > > (Frantar & Alistarh, 2023) Frantar, E., & Alistarh, D. (2023, July). Sparsegpt: Massive language models can be accurately pruned in one-shot. In International Conference on Machine Learning (pp. 10323-10337). PMLR.
> > > > (Sun et al., 2023) Sun, M., Liu, Z., Bair, A., & Kolter, J. Z. (2023). A simple and effective pruning approach for large language models. arXiv preprint arXiv:2306.11695.
> > > > ```
> > > >
> > > > ### R2
> > > >
> > > > > **For W2**, you mentioned,
> > > > *"We will include training time in the experiment in our revision."*
> > > > Could you please upload your revision or share the results in the review thread here?
> > > >
> > > > The actual training times of the algorithms involved in our experiment are same because these algorithms perform dense matrix operations on the actual hardware. However, it is important to emphasize that our contribution is at the algorithmic level, and corresponding hardware support is beyond the scope of our work. We will include the specific training times of these algorithms in the revision.
> > > >
> > > > ### R3
> > > >
> > > > > **For W4**, my question,
> > > > *"how can Hybrid Sparse Attention be combined with FlashAttention to increase FLOPs/s? Claims of MST’s compatibility with other accelerations, such as training parallelism and hardware-assisted methods, lack demonstration. Specific evaluations, such as integrating Hybrid Sparse Attention with FlashAttention, could strengthen this claim"*
> > > > has not been answered.
> > > >
> > > > It may be possible to combine, as our method is an algorithmic-level acceleration, which is on a different level from other hardware acceleration methods. As mentioned in our previous response, Hybrid Sparse Attention is an optimization at the algorithmic level, and in principle, it can be directly compatible with hardware acceleration methods like FlashAttention. Specifically, FlashAttention improves the speed of attention computation by using blocking, while HSA also sparsely blocks the matrices involved in attention.
> > > >
> > > > ### R4
> > > >
> > > > > **For W3**, while I understand your resource limitations, experimenting with a 1B LLaMA or a 3B model seems feasible and would strengthen your work.
> > > >
> > > > Thank you for your understanding. We will try our best to conduct experiments on a 1B LLaMA or a 3B model in our revision. However, we believe that our work will have a broad impact on the LLM community, making it possible for larger models to achieve considerable savings during pre-training.
> > > >
> > > > ---
> > > > We are grateful for your constructive suggestions, which have significantly guided our improvements. We hope our response addresses your concerns. We will also be happy to answer any further questions you may have. Thank you very much!

---

### Decision · Action_Editor_66i7 · 2025-02-14

**Recommendation:** Accept with minor revision

**Comment:**

As above.

**Audience:**

Yes. With the new results on wall clock speedup, I believe this will be a useful paper for the TMLR audience.

**Claims And Evidence:**

This paper aims to speed up transformer training by using mixed sparsity training (MST). The authors observe that transformers exhibit considerable redundancy in pretraining computations, which motivated their solution MST. The authors observe a FLOP reduction of 4x without compromising performance.

Generally, this is a valuable contribution. However, two reviewers still have lingering concerns regarding the lack of reporting of wall clock speedup and consistent results on larger models. I will recommend acceptance with minor revision. The authors should report the wall clock speedup and I will look through the revised manuscript before making my final decision.

---

> ### Author Response · Authors · 2025-03-10
> **Response to Action Editor 66i7**
>
> Thank you for your time and effort in handling our paper, and for recognizing the value of our work! The wall-clock time has been included in *Appendix B.1* of our camera-ready version. Due to hardware constraints, all tensor computations in our experiments were performed using dense methods, leading to similar wall-clock times for sparse algorithms, including MST. However, this does not diminish the significance of our contributions.
>
> We would like to emphasize that speedups at the algorithmic level and corresponding hardware support do not always progress in sync. For example, sparsification for training neural networks was first proposed and studied at the algorithmic level (Li et al., 2017; Frankle et al., 2019), while the first widely adopted hardware architecture supporting structured sparsity was implemented by NVIDIA in 2020 (NVIDIA, 2020). As a result, the actual training times for the algorithms in our experiment are unlikely to differ significantly, as these algorithms primarily involve dense matrix operations on the hardware. Our work introduces a novel optimization approach at the algorithmic level, which we believe can inspire further advancements at the hardware level.
>
> ```
> (Li et al., 2017) Hao Li, Asim Kadav, Igor Durdanovic, Hanan Samet and Hans Peter Graf. Pruning Filters for Efficient ConvNets (ICLR 2017)
> (Frankle et al., 2019) Jonathan Frankle and Michael Carbin. The lottery ticket hypothesis: Training pruned neural networks (ICLR 2019)
> (NVIDIA, 2020) NVIDIA. https://developer.nvidia.com/blog/nvidia-ampere-architecture-in-depth/ (2020)
> ```